# Towards Stability of Autoregressive Neural Operators

**Michael McCabe**[*]                                                                  *michael.mccabe@colorado.edu*
*Department of Computer Science*
*University of Colorado Boulder*

**Peter Harrington**[†]                                                                  *pharrington@lbl.gov*
*Lawrence Berkeley National Laboratory*

**Shashank Subramanian**[†]                                                          *shashanksubramanian@lbl.gov*
*Lawrence Berkeley National Laboratory*

**Jed Brown**                                                                            *jed@jedbrown.org*
*Department of Computer Science*
*University of Colorado Boulder*

Reviewed on OpenReview: *https://openreview.net/forum?id=RFfUUtKYOG*

## Abstract

Neural operators have proven to be a promising approach for modeling spatiotemporal systems in the physical sciences. However, training these models for large systems can be quite challenging as they incur significant computational and memory expense—these systems are often forced to rely on autoregressive time-stepping of the neural network to predict future temporal states. While this is effective in managing costs, it can lead to uncontrolled error growth over time and eventual instability. We analyze the sources of this autoregressive error growth using prototypical neural operator models for physical systems and explore ways to mitigate it. We introduce architectural and application-specific improvements that allow for careful control of instability-inducing operations within these models without inflating the compute/memory expense. We present results on several scientific systems that include Navier-Stokes fluid flow, rotating shallow water, and a high-resolution global weather forecasting system. We demonstrate that applying our design principles to neural operators leads to significantly lower errors for long-term forecasts as well as longer time horizons without qualitative signs of divergence compared to the original models for these systems. We open-source our code for reproducibility.

## 1 Introduction

There is an increasing interest in using neural networks to simulate scientific systems described by partial differential equations (PDEs), with a specific focus on *neural operators*. These are neural network models that learn the mapping between two function spaces that represent the inputs and solutions to a PDE through training using a finite collection of input-solution pairs of data. (Li et al., 2021b; Lu et al., 2021, and references therein). These models have demonstrated great success in simulating PDE systems across a broad range of scientific disciplines. With the success of large neural network models such as foundation models (Bommasani et al., 2021; Devlin et al., 2018) in natural language processing (NLP) and computer vision (CV), recent efforts have attempted to scale these neural operators for grand challenge problems in scientific computing, particularly in geophysical systems like weather and climate. These methods have demonstrated impressive performance in predicting crucial physical variables and are comparable to traditional numerical methods over short time horizons under certain metrics, while enjoying orders of magnitude improvement in computational

---

[*]This work was conducted in part while author was intern at Lawrence Berkeley National Laboratory.
[†]Equal contribution.

expense in an operational (inference) setting. For instance, in weather forecasting, many models (Pathak et al., 2022; Bi et al., 2022; Lam et al., 2022) are able to achieve (or even beat) the deterministic accuracy of numerical methods with 1000× faster times-to-solution.

Scaling neural operators to these domains, however, presents unique challenges. First, even at modestly high resolutions (such as a 25 km grid over the earth for geophysical systems), a single physical state variable, represented as an image channel, consists of more than a million grid points (pixels). This is about *20×* *larger* than standard CV image datasets such as ImageNet (Deng et al., 2009). For some systems, there may be hundreds of such state variables represented at each grid point (e.g., Hersbach et al., 2020). The high dimensionality places strict limitations on architectures that scale poorly with resolution and/or channel dimensions.

To account for scaling difficulties while making spatiotemporal predictions, most models predict temporal sequences of physical states in an *autoregressive* manner. Prior efforts across a wide set of spatiotemporal forecasting problems have observed an inherent trade-off in autoregressive forecasting: smaller time-steps typically result in an easier one-step task, but total error grows with an increasing number of steps even over a fixed time window (Bi et al., 2022; Tran et al., 2023; Li et al., 2021a; Stachenfeld et al., 2022; Krishnapriyan et al., 2022). Methods to reduce this autoregressive error growth to enable either longer forecasts or smaller step sizes have included using multiple models trained at different timescales to reduce the total number of steps (Bi et al., 2022), learned step sizes (Tran et al., 2023), dissipative priors (Li et al., 2021a), spectral disretization in time (Li et al., 2021b), noise injection (Stachenfeld et al., 2022), and multi-step rollouts during training (Lam et al., 2022; Pathak et al., 2022). These approaches have shown promise but many drastically increase training cost (multi-step, multi-model, spectral-in-time), add additional hyperparameters that must be tuned (noise injection), or are only applicable in certain cases (dissipative priors).

In this paper, we critically analyze sources of this autoregressive error growth. We highlight geophysical applications as earth systems are both large in scale and often demand long-range forecasts of physical state variables—for example, forecasts of the atmospheric state (wind, temperature, precipitation, etc.) are commonly run for several years to understand the atmosphere's global behavior, particularly with a changing climate. Autoregressive error corruption renders such forecasts inadequate for any scientific analyses and it is critical to develop methods to mitigate this. We focus on models derived from prototypical neural operators such as variants of the Fourier Neural Operator (FNO) (Li et al., 2021b) which have demonstrated promising results on various scientific applications that include earth systems (Pathak et al., 2022). These models utilize spectral transformations to implement global operations in a compute-efficient manner—for example, the FNO implements global convolutions through fast Fourier transforms (FFTs). We suggest novel changes to their model architectures by borrowing ideas from classical numerical methods such as pseudospectral schemes for solving nonlinear time-dependent PDEs as well as application-specific modifications that address instabilities that arise due to the structure of the problem.

Specifically, our contributions are as follows:

1. **Connecting sources of instability and numerical analysis.** We draw parallels between instabilities that arise in autoregressive spatiotemporal forecasting of physical state fields and standard numerical analysis through the lens of pseudospectral numerical schemes and demonstrate that autoregressive spatiotemporal models show signs of aliasing and numerical instability, contributing to nonlinear error growth and divergence (much like in pseudospectral numerical methods).

2. **Controlling error growth through model and application-specific innovations.** We introduce several modifications to our architectures to control the above error growth without inflating our computational/memory expense. These include (1) frequency-domain spectral normalization to control the sensitivity of spectral convolutions, (2) depthwise-separable spectral convolutions for efficient parameter usage, (3) adapting the classical Double Fourier Sphere (DFS) method to represent functions defined on a spherical geometry as a torus for geophysical applications, and (4) data-driven spectral convolution.

3. **Boosting long-range autoregressive performance.** We demonstrate the efficacy of our proposals by applying them in off-the-shelf fashion to standard neural operators such as the FNO and its variants. Our experiments across multiple physical systems including Navier-Stokes benchmark

fluid simulations, shallow water equations on rotating sphere, and weather forecasting demonstrate improved stability and long-term accuracy. The last system is an especially challenging problem that models the high resolution spatiotemporal patterns of atmospheric variables on a planetary scale with broad scientific and societal implications. In this setting, even for particularly volatile fields such as surface wind, we show that our improvements enable up to 800% longer instability-free forecasts, paving the way for long-range predictions using neural operators. Models for smoother problems like shallow water using our modifications show no signs of instability at any point in our experiments.

*Outline.* We first provide some background on the connection between pseudospectral methods and our observed sources of instability in neural operator models in §2. We then explore these sources of error in the context of neural operators in §3, introduce our innovations to overcome these in §4, and finally demonstrate the impact of our innovations across different physical systems in §5.

## 2 Background

### 2.1 Fourier Pseudospectral Method

The standard approach for introducing neural operators explores the connection between neural operator methods and integral methods for the solution of linear PDEs (Li et al., 2021b). However, these methods also have a strong mechanistic similarity to pseudospectral methods used for solving time-dependent nonlinear PDEs (Orszag, 1970; Eliasen et al., 1970)—we use this alternative perspective to motivate their construction as the connection exposes sources of numerical instability in these models.

Spectral methods approximate a function $u(x, t)$ defined over space and time by a linear combination of $K$ basis functions $u_{|K}(x, t) = \sum_{k=0}^{K} U_k(t)\phi_k(x)$ where $\phi_k$ is the $k$th element of the set of basis functions and $U_k(t)$ is the corresponding time dependent coefficient. Bases are chosen to allow for efficient numerical algorithms while offering rapidly decreasing truncation error $T_{|K}[u]^2(t) = \sum_{k=K}^{\infty} U_k^2(t)$ as $K$ increases. A function that can be represented with no truncation error for finite $K$ is called *bandlimited*. In this work, unless explicitly stated otherwise, we will be referring to the Fourier basis $\phi_k(x) = e^{i2\pi kx}$. We note that this choice of basis implicitly assumes periodic boundary conditions which will be discussed later in the context of earth systems. Pseudospectral methods differ from pure spectral methods in that they augment the spectral representation with a representation in the spatial domain sampled at $N$ points $\{x_1, \ldots, x_N\}$ (Fornberg, 1996) such that now both our spectral and spatial representations consist of discrete sets of elements. This property, along with the existence of fast spectral transforms, enables efficient computation of nonlinear terms.

Relying on discrete transforms introduces complications that must be accounted for in pseudospectral methods. Instead of computing true basis coefficients $U_k(t) = \int_\Omega u(x, t)\phi_k(x) \, dx$, we instead compute the finite approximation $\hat{u}_k(t) = \sum_{n=1}^{N} \phi_k(x_n)u_n(t)$. Different basis functions may be indistinguishable when sampled at a finite number of points and so the estimated coefficients become the sum of the true coefficients of these indistinguishable basis functions. This phenomenon is called *aliasing* and the unresolvable modes are called *aliases*. For the Discrete Fourier Transform (DFT), modes above the Nyquist frequency of $N/2$ are aliases of lower frequency modes (Smith, 2007). The discrete aliasing error for the DFT can be defined as $A_{|K}[u(t)]^2 = \sum_{k=-K}^{K} (\sum_{j \neq 0} U_{k+jN}(t))^2$. Aliasing is often introduced by the application of *nonlinear* transformations applied in the spatial domain through a process described below.

To explore this, we examine the following one-dimensional nonlinear advection equation applied to a time-varying scalar field $u \in \mathcal{L}^2(\mathbb{T})$ defined on the periodic domain $x \in [0, 1]$:

$$\frac{\partial u(x, t)}{\partial t} = -u(x, t)\frac{\partial u(x, t)}{\partial x} \tag{1}$$

Assuming that $u$ has bandlimit $K = \frac{N}{2}$, we can exactly represent $u$ by the Fourier basis coeffiecients $\hat{u}_k$ computed by the DFT. In frequency space, applying linear differential operators can be done via element-wise multiplication and applying 1 amounts to reducing the equation to a system of ODEs defined by the

convolution:

$$\frac{\partial \hat{u}_k(t)}{\partial t} = \left( \hat{u}(k,t) * \widehat{\frac{\partial u}{\partial x}}(k,t) \right)[k], \quad \forall k \in \{-2K, \ldots 2K\}. \tag{2}$$

Computing these derivatives is where the pseudospectral method has a significant advantage over spectral methods. The direct convolution in the frequency domain has a cost of $O(N^2)$, but by the dual of the Convolution Theorem (Smith, 2007), we could also compute this by transforming back into the spatial domain ($O(N \log N)$) and subsequently performing elementwise multiplication ($O(N)$). Once the time derivatives are computed, one can use any off-the-shelf ODE integrator to march the equation forward via the method of lines (Boyd, 2013).

While the transform approach is significantly more efficient, computing the nonlinear term in the spatial domain introduces aliasing. The bandlimit of the newly computed time derivative is up to twice that of the original function (for the full derivation, see Appendix A.1). Without increasing the spatial sampling rate, these newly generated modes will alias back onto modes below the Nyquist limit. This is true not only for polynomials as seen here, but also many commonly used nonlinear functions in deep learning, which can be seen informally through the Weirstrass Approximation Theorem since our continuous nonlinearities can be arbitrarily closely approximately by infinite series of polynomials over a finite interval (Hunter & Nachtergaele, 2001). Aliasing is a major contributor to instability for pseudospectral methods and, as we will see in later sections, in the use of autoregressive neural operators as well.

## 2.2 Connection to Fourier Neural Operators

The pseudospectral method can be summarized as using fast transforms to alternate between the spectral domain, where we can quickly compute high-accuracy spatial derivatives via convolution, and the spatial domain where we can compute nonlinearities with similar efficiency. The resulting value is then added back to the original state according to an ODE integration rule. We note a similar pattern in Fourier Neural Operators (FNO).

Consider one FNO block applied to a problem with one spatial dimension. Let $v^{(\ell)} \in \mathbb{R}^{N \times D^{(\ell)}}$ denote the hidden state at layer $\ell \in \{0, \ldots, L\}$ at each discretization point such that $v^{(0)} = u$. We use $N, K, D$ to denote the sizes of the spatial, frequency, and channel dimensions respectively. The block is parameterized by $W^{(\ell)} \in \mathbb{R}^{D^{(\ell+1)} \times D^{(\ell)}}$ and $R^{(\ell)} \in \mathbb{C}^{K \times D^{(\ell+1)} \times D^{(\ell)}}$ and computed as:

$$v_{ic}^{(\ell+1)} = h\left( \sum_{c'=1}^{D^{(\ell)}} W_{cc'}^{(\ell)} v_{ic'}^{(\ell)} + \left[ \mathcal{F}^{-1} \mathcal{K}^{FNO}(\mathcal{F} v^{(\ell)}) \right]_{ic} \right) \tag{3}$$

$$\mathcal{K}^{FNO}(\mathcal{F} v^{(\ell)})_{kc} = \sum_{c'=1}^{D^{(\ell)}} R_{kcc'}^{(\ell)} [\mathcal{F} v^{(\ell)}]_{kc'} \tag{4}$$

where $\mathcal{F}$ denotes the Discrete Fourier Transform and $h$ denotes an arbitary nonlinear activation function. Note that we use $i, k$ to index the spatial/frequency domains respectively. In subsequent sections we will refer to the first term inside the nonlinearity of Equation 3 as a pointwise linear operation (or equivalently 1x1 convolution) and the second term, which is expanded in Equation 4, as the spectral convolution.

For the purposes of our later discussion on aliasing mitigation, the key connection between the FNO and pseudospectral methods is the fact that both methods repeatedly alternate between the spectral domain where they compute large-kernel convolutions and the spatial domain where they apply nonlinearities. In both cases, the nonlinearity in the spatial domain potentially introduces aliasing while the spectral transformation offers opportunities for efficiently mitigating this aliasing. This connection allows us to explore solutions from the pseudospectral literature for stability problems faced by the FNO. Deeper connections in terms of functional form are derived for interested readers in Appendix A.2.

Section 3.1 empirically explores the consequences of this shared aliasing behavior while Section 4.2 proposes mitigation strategies.

## 2.3 Related Work

**Aliasing in Deep Learning.** While aliasing in deep neural networks has previously been studied in classification and generative models Zhang (2019); Zou et al. (2020); Vasconcelos et al. (2021); Ribeiro & Schön (2021); Karras et al. (2021), there has been a surprisingly small amount of exploration in the context of spatiotemporal forecasting. Zhang (2019) previously showed that aliasing allows convolutional architectures to learn non-shift equivariant features. Zou et al. (2020), Ribeiro & Schön (2021), and Vasconcelos et al. (2021) demonstrate in multiple settings that while low-pass filtering is valuable to improving model accuracy, CNNs typically do not allocate capacity to do so via normal training procedures. In the generative space, Karras et al. (2021) observed significant improvements to consistency in GAN-generated data by employing anti-aliasing measures. Within the context of neural operators, Fanaskov & Oseledets (2022) showed that aliasing caused by ReLU nonlinearities can lead to significant error. Their fully spectral solution avoids aliasing, but scales poorly due to $O(K^2D)$ complexity in frequency mode mixing.

**Fourier Neural Operators.** Fourier neural operators and their variants (Li et al., 2021b) have been a popular and successful class of neural networks used for solving PDEs. They have been shown to be efficient universal approximators for the solution of PDE systems (Kovachki et al., 2021; De Ryck & Mishra, 2022) and have been successfully applied to scientific computing problems in a variety of challenging domains (Pathak et al., 2022; Wen et al., 2022; Li et al., 2022; Guibas et al., 2021; Guan et al., 2021; Yin et al., 2022; Witte et al., 2022; Li et al., 2021a; Tran et al., 2023; Fanaskov & Oseledets, 2022), particularly those related to fluid dynamics and earth systems. Our focus in this paper is on stabilizing and improving the autoregressive use of neural operators for spatiotemporal problems (also referred to as Markov Neural Operators in Li et al., 2021a), which is a necessity for applying these methods to large-scale systems. The Markov Neural Operator of Li et al. (2021a) is shown to reproduce statistical properties of the attractor in dissipative chaotic dynamical systems, allowing long-term autoregressive forecasting via use of Sobolev losses and dissipative data augmentation. Here, we also aim to achieve stable trajectories with autoregressive FNOs, but target systems with external, non-constant forcings and focus on architecture-specific sources of error. While we showcase the FNO architecture and its variants due to their connection with pseudospectral PDE methods, many of our contributions can be applied to any neural network used for autoregressive temporal forecasting.

**Spherical Architectures.** One motivation for addressing stability in a scalable manner is to apply these tools to earth systems. In these settings, we must also account for spherical geometry. We address this in a lightweight manner in FNO via use of the DFS method, but other neural architectures designed for spherical geometry (Cohen et al., 2018; Esteves et al., 2017; Defferrard et al., 2020) are relevant to our discussions, and we directly compare our model against some of these in Section 5.1. We note that in order to to make the cost of handling spherical geometry minimal, our DFS approach does not seek to obtain exact $SO(3)$ equivariance as found in the $SO(3)$ or spherical convolutions of of Cohen et al. (2018) or Esteves et al. (2017).

# 3 Where Autoregressive Neural Operators Fail

## 3.1 Aliasing and Unbounded Out-of-Distribution Error Growth

We begin by demonstrating experimentally where this autoregressive growth occurs in the spectrum. For this, we use a simplified version of the Navier-Stokes problem with Kolmogorov forcing from Kochkov et al. (2021) — this system is regularly used for testing neural operator models. Usually, the objective is to predict a single timestep significantly larger than the timestep of the integrator used to generate the data. The large timestep is the key to the speed-up offered by neural operator methods over conventional numerical methods. However, our goal in using this system is to demonstrate the growth of autoregressive error on a nontrivial task in which prior work has established the feasibility of approximating dynamics over the given interval, so we instead predict the result of a $300\times$ smaller step to produce what should be an even easier problem than the original test case. Architectural and training details for figures related to this experiment can be found in Appendix C.1.

Figure 1 shows that the vanilla FNO must sacrifice resolution for stability. While the gap is marginal, for the initial steps, the FNO with no spectral truncation outperforms its truncated relatives, but the FNO without spectral-in-time discretization diverges even on this relatively simple task within a small number of

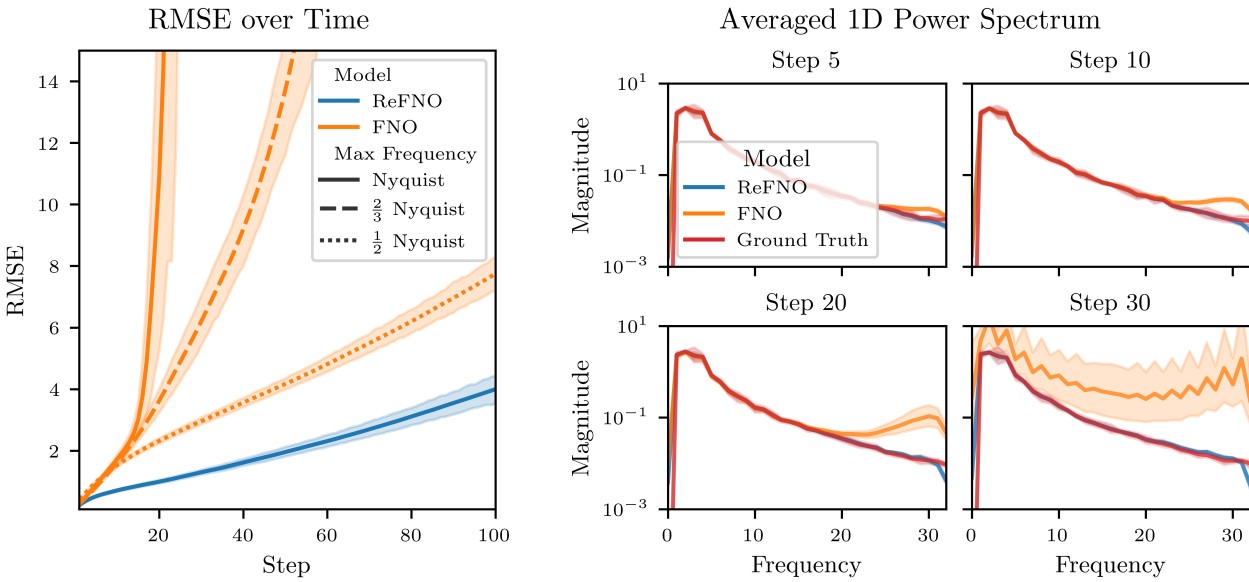

Figure 1: Navier-Stokes Error and Spectra: (Left) Conventional spectral neural operators such as the FNO can gain short-term accuracy by using more modes at the cost of long run stability while our restructured ("Re" prefixed) approach is able to obtain the best of both worlds. (Right) The instability in the full spectrum models starts in the high frequencies and grows. This is consistent with aliasing behavior.

steps. The right-hand side which depicts the spectra of un-truncated models begins to give us some insight into what is happening here. For these full spectrum models, we observe uncontrolled error growth in the high frequencies. This behavior is consistent with the accumulation of aliasing error. Recall from Section 2.1, that for $N$ samples with truncation limit $K$, the discrete aliasing error is defined as:

$$A_{|K}[u(x,t)]^2 = \sum_{k=-K}^{K} \left(\sum_{j\neq 0} U_{k+jN}(t)\right)^2 \tag{5}$$

The conjugate symmetry of the spectrum of real-valued functions implies that the magnitude of newly generated frequencies immediately above the bandlimit fold back onto frequencies immediately below the truncation limit causing the observed pileup near the Nyquist frequency *.

These issues are not limited to simple or small-scale tasks. Figure 2 shows autoregressive rollouts from FourCastNet (FCN, Pathak et al., 2022), a model based on the Adaptive Fourier Neural Operator (Guibas et al., 2021) used for high-resolution autoregressive global weather forecasting, and also accumulates spectral pathologies. Here we can observe a similar buildup, though in slightly different fashion as the spikes now occur near the aliases of the zero frequency. The differences in the spectral pathologies between isotropic and multi-resolution architectures are explored briefly in A.3 though in a limited sense and we believe it is an interesting topic for future work. As errors from aliasing or other sources of inaccuracy accumulate during autoregressive time-stepping, the inputs passed to a model will gradually drift away from the training distribution into spaces where the numerical behavior of the trained model is difficult to characterize. Thus in addition to mitigating aliasing, in Section 4.1, we explore approaches to model less sensitive to small movements away from the training distribution in a scaleable manner.

## 3.2 Geometry and the Fourier Transform

We highlight geophysical applications as they have seen some of the largest scale uses of neural operators. Our goals include being capable of scaling our contributions to the magnitude of earth systems. The FNO does not

---

*This is often called "spectral blocking" (Boyd, 2013) as the truncation limit effectively blocks the movement of energy into higher wave numbers causing pileup near the Nyquist frequency.

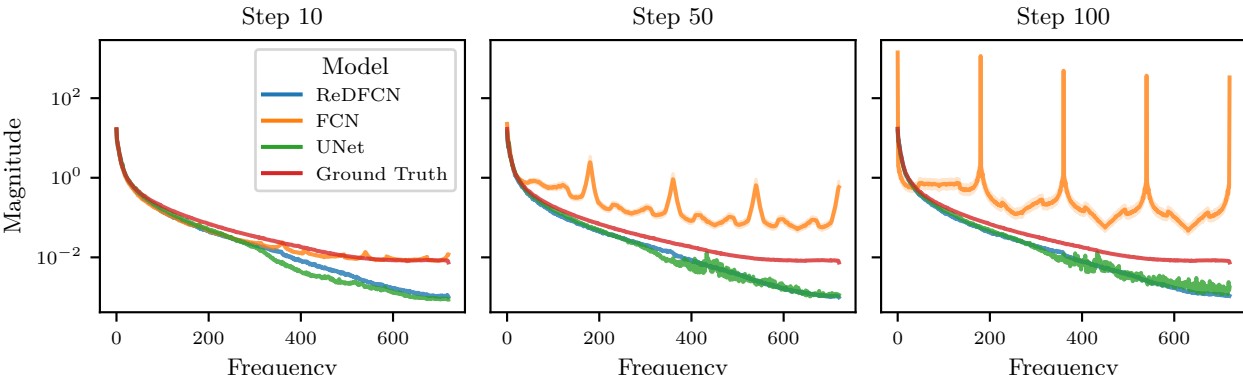

Figure 2: Averaged 1D Spectrum for FourCastNet (FCN): FCN begins demonstrating spikes in the spectrum corresponding to aliases of the zero frequency while our restructured version (ReDFCN) has consistent behavior over time. Convolutional baselines start off strong but eventually develop noisy spectra from poor geometric handling.

inherently scale to these problem sizes, but the AFNO, a more parameter-efficient variant, was successfully used in FourCastNet (FCN), one of the first neural network models to demonstrate strong performance on high-resolution global weather forecasting. However, the 2D FFT used in this model for spectral convolution proves to be a problem in the spherical setting. It implicitly considers the globe as periodic both in the zonal (east-west) direction and in the meridional (north-south) directions. This representation leads to an artificial discontinuity and allows for "local" operators to exchange information across the poles. Moreover, representing discontinuous functions in the Fourier basis can lead to spurious maxima through the Gibbs phenomenon (Boyd, 2013). Some downstream effects of this pathology are depicted in Figure 4, showing model Jacobians "looking" pole-to-pole (panel b) which can lead to development of unphysical artifacts at the poles (panel c). These can grow and propagate to the rest of the domain, destabilizing model forecasts.

## 4 Mitigating Known Sources of Error

In this section, we list our architectural improvements which directly address the aforementioned issues relating to aliasing, sensitivity, and geometry, as well as other orthogonal innovations that improve efficiency and scalability of the underlying architecture.

### 4.1 Efficiently Reducing Model Sensitivity

From the lens of pseudospectral methods, the spectral operations in the FNO are seen as a simple spectral-domain convolution (Rippel et al., 2015) rather than as the realization of an integral equation solver. This viewpoint gives us the flexibility to borrow tools developed for general convolutional networks and adapt them to the spectral convolution setting which is what we do here.

**Depthwise Separable Spectral Convolution.** Popularized by the Xception architecture (Chollet, 2017) and MobileNet (Howard et al., 2017), depthwise separable convolutions decouple spatial convolution from channel mixing to trade expressive power at a given width for memory savings via a parameter count reduction. In most cases, this savings is then spent to expand the width of a layer. As the rapid parameter growth rate of the FNO is one of the leading obstacles to scaling, creating a depthwise separable variant of spectral-domain convolution is a natural approach to improve scalability.

Recall from Equation 4 that the FNO employs dense spectral convolutions with the form:

$$\mathcal{K}^{FNO}(\mathcal{F}v^{(\ell)})_{kc} = \sum_{c'=1}^{d^{(\ell)}} R_{kcc'}^{(\ell)}[\mathcal{F}v^{(\ell)}]_{kc'} \tag{6}$$

where $R^{(\ell)} \in \mathbb{C}^{K \times D^{(\ell+1)} \times D^{(\ell)}}$. We can define a depthwise separable convolution in the spectral domain by instead parameterizing the convolution with a decoupled channel mixing matrix $V \in \mathbb{R}^{d^{(\ell+1)} \times d^{(\ell)}}$ and spectral filter $r^{(\ell)} \in \mathbb{C}^{K \times d^{(\ell)}}$ and computing:

$$\mathcal{K}^{DS}(\mathcal{F}v^{(\ell)})_{kc} = \sum_{c'=1}^{d^{(\ell)}} V_{cc'}^{(\ell)} r_{kc'}^{(\ell)} [\mathcal{F}v^{(\ell)}]_{kc'}. \tag{7}$$

Note that the channel mixing parameterized by $V$ can be performed in the spatial domain for a small efficiency savings. This reparameterization reduces the parameter growth rate from $N \times D^{in} \times D^{out}$ to $(N \times D^{in}) + (D^{in} \times D^{out})$ which in modern architectures can reduce parameter counts by hundreds or close to a thousand times. It is possible to reduce this even further for larger problems through the use of meta-networks similar to those used for continuous position embeddings (Liu et al., 2022a) to generate convolutional filter weights, though at the cost of additional operations.

**Spectral Normalization.** In Figure 1, we see that once the data becomes sufficiently corrupted, error growth becomes exponential. This behavior may be partly explained by examining the distribution of singular values in weight matrices. Empirically, we find that singular values can be very large, up to 50 for some layers. This sensitivity concern is a well-known issue (Liu et al., 2020; Van Amersfoort et al., 2020; Rosca et al., 2021) in other tasks, particularly uncertainty quantification. One mitigation approach is spectral normalization (Miyato et al., 2018), which ensures that behavior both in and out-of-distribution during training is bounded by rescaling weights by the spectral norm (or equivalently the maximum singular value) of the weight matrix. This ensures that the largest singular value of the weight matrix is less than one.

Circulant matrices, which represent discrete circular convolution in the spatial domain, are well known to be diagonalized by the Discrete Fourier Transform with eigenvalues equal to the Fourier coefficients of the convolutional filter (Smith, 2007). In the case of spectral convolution, the filter is already parameterized in frequency space, so we can adapt the spectral normalization constraint exactly and differentiably by applying a squashing function to the polar parameterization of the filter coefficients. Representing the filter coefficients $r_k^{(\ell)}$ in polar form $r(a, \theta) = a e^{2\pi i \theta}$, it is sufficient to constrain $|a| \leq 1$ which can be achieved by constructing the spectral filter coefficients as:

$$r(a, \theta) = \sigma(a) e^{i\theta} \tag{8}$$

where $\sigma$ is the sigmoid nonlinearity. This ensures that the spectral norm of the convolution operator is bounded from above by one. We use this procedure for spectral domain spatial convolution and traditional spectral normalization for pointwise linear operations.

## 4.2 Aliasing and Learning to Filter

One of the most prevalent approaches for eliminating aliasing in pseudospectral methods is oversampling in the spatial domain or, equivalently, filtering a fixed number of the highest frequencies in the Fourier domain (Orszag, 1971). The polynomial nonlinearities frequently encountered in nonlinear PDEs produce new Fourier modes in a predictable manner, so it is often possible to define explicit oversampling or low-pass filtering strategies which exactly eliminate aliasing depending on the order of the polynomial. These schemes can significantly increase the cost of the simulation, but are often necessary for accurate, stable computation.

The frequency-domain behavior of the non-polynomial nonlinear functions used in deep learning is significantly more complicated and can vary depending on input values — for analytic functions, this can be directly observed through analysis of the Taylor expansion around different points. In these cases, a fixed truncation strategy may be either computationally wasteful and overly diminish frequency-domain resolution or may be insufficient depending on the problem. The spectral convolution of the FNO can learn problem-specific truncation strategies by setting filter coefficients at or very close to zero; however, this is not enough to perform low-pass filtering in the conventional FNO block. As we can see in panel (a) of Figure

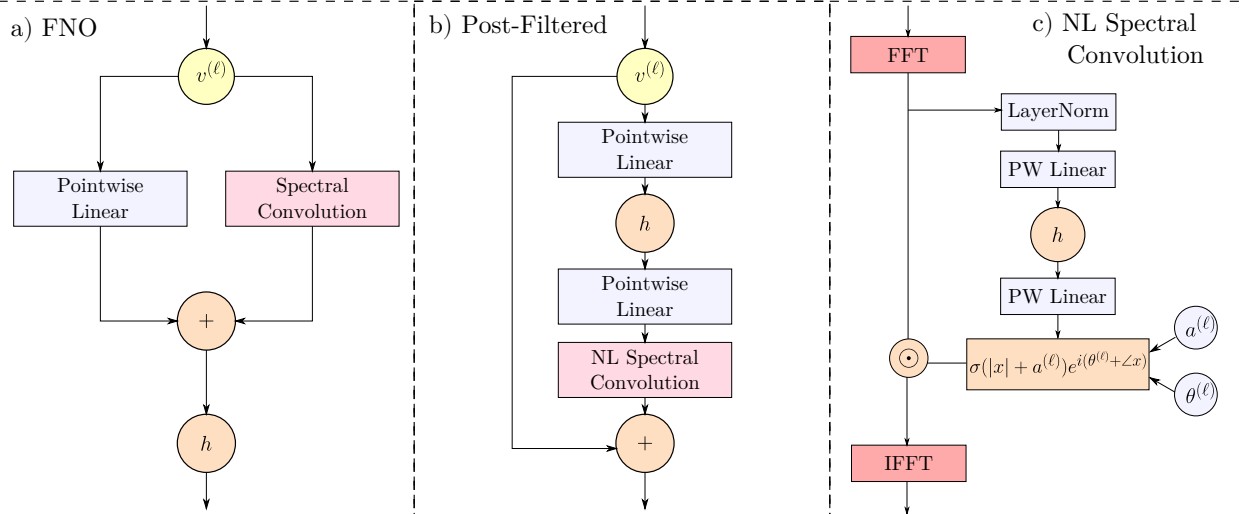

Figure 3: a) In the FNO, the aliasing-inducing nonlinearity ($h$) is applied to both paths, including the path that skips the spectral convolution. This makes it difficult for the model to learn to deal with these newly generated high frequency modes as there is always a path that is unfiltered. b) By ensuring that all data passing through the nonlinearity also passes through the spectral convolution, we allow the network the opportunity to learn to filter any new frequency modes. c) Since the aliasing patterns of non-polynomial nonlinearities are data dependent, we further augment this approach with a nonlinear spectral convolution which learns data-dependent offsets ($|x|, \angle x$) for learned filter coefficient components ($a, \theta$) via mode-wise MLP.

3, the standard FNO structure includes pointwise linear operations which bypass the convolution but still pass through the nonlinearity. Due to this, learning a simple low-pass filter would actually require exact cancellation of modes by the fixed nonlinearity.

We propose to restructure the FNO block so that nonlinearities are always followed by learnable filters capable of directly filtering newly generated high frequencies. Furthermore, since the aliasing behavior of the nonlinearities used in deep learning can vary depending on the input, we additionally replace the static convolutional filter of the FNO with one which is a dynamic function of the input data. This proposed block is presented in panels (b) and (c) of Figure 3. Dynamically generated filters would be infeasible in the dense convolution case where we would need to predict $D^2$ weights for each mode, but in depthwise separable spectral convolution, the convolutional filter $r$ has the same dimensionality as the input data, allowing us to generate filters by a simple mode-wise MLP in the frequency domain. These dynamic filters are implemented as data-driven offsets to learned biases in both the phase ($\theta$) and magnitude ($a$) components of the filter. The data-driven and learned components are then combined and the magnitude component is squashed by a sigmoid activation as described in Section 4.1. We note that while we are now using a nonlinear function to generate the filters, the application of the filters remains a linear operation and does not produce new modes.

## 4.3   Improved Domain Representation

One simple approach which can correct the artificial discontinuity induced by the 2D FFT while requiring almost no architectural modifications is the Double Fourier Sphere (DFS) method (Merilees, 1973; Orszag, 1974). Given an equirectangular grid defined by uniform spacing in spherical coordinates $\lambda \in [0, 2\pi], \theta \in [0, \pi]$ (longitude, colatitude), DFS converts the original zonally periodic domain into a new domain defined on a torus. This is done by concatenating the original data with its half-phase glide reflection (Martin, 1996), a transformation consisting of a reflection followed by a translation of that reflection such that a column in the north-south direction of the concatenated representation follows a line of longitude around the sphere.

The mathematical formulation can be found in Appendix A.5. A sample visualization of this process for an atmospheric wind snapshot is given in Figure 4(a).

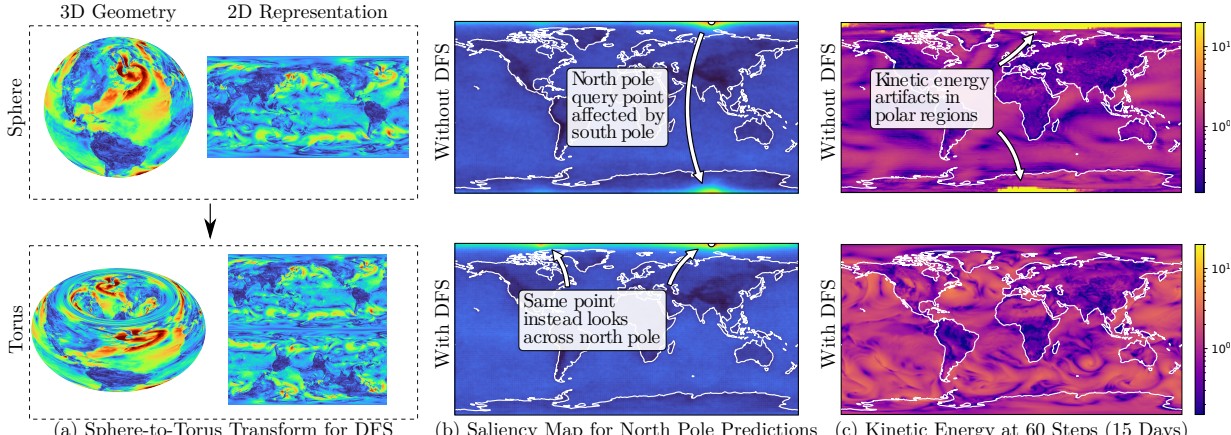

(a) Sphere-to-Torus Transform for DFS    (b) Saliency Map for North Pole Predictions    (c) Kinetic Energy at 60 Steps (15 Days)

Figure 4: (a) The sphere-to-torus transformation of Merilees (1973) for the Double Fourier Sphere method. (b) Multi-sample averaged absolute value of derivative of predicted v10 value at a north pole pixel with respect to all spatial locations from FourCastNet trained with and without the DFS transformation. The original formulation leads to the prediction at the North pole showing high sensitivity to input values at south pole. (c) Predicted surface kinetic energy after 60 autoregressive steps for FCN models trained with and without DFS. Without DFS, the polar discontinuity can contribute to rapid divergence near the poles. The Double Fourier Sphere transform is less widely used than alternative bases such as the spherical harmonics (Atkinson & Han, 2012), but offers advantages in both computational and implementation complexity as it amounts to structured padding followed by a 2D FFT. Furthermore, the DFS mapping has been proven to be continuous for certain function classes (Mildenberger & Quellmalz, 2022), has shown use in fast low-rank approximation (Townsend et al., 2015), and numerical experiments using the basis have obtained similar accuracy to full spherical harmonic models (Spotz et al., 1998; Cheong, 2000; Park et al., 2013; Fornberg, 2006). In Figure 4, we can see that this relatively simple operation is sufficient to eliminate the nonphysical cross-polar behavior in FourCastNet as the saliency map (Simonyan et al., 2014) no longer shows cross-polar communication and subsequently the polar growth is eliminated.

While DFS resolves the artificial discontinuity introduced by the 2D FFT, naive implementations introduce two distinct issues which we introduce further refinements to address below:

**Shaping with Geometry-Aware Filters.** Spherical harmonics with the conventional triangular truncation are isotropic in the sense that their resolution does not vary over the sphere. On the other hand, the DFS transform applied to a uniform grid has significantly higher resolution near the poles than at the equator which can lead to spatial distortion. This distortion, however, can be limited by filtering the frequency domain according to Spotz et al. (1998), who found that applying low-pass filters on a latitude-by-latitude basis can mimic the isotropy of spherical harmonics and greatly improve the stability of DFS-based numerical simulations. To apply latitude-dependent filters for a given colatitude $\theta$, we note that the wavelength of zonal wavenumber $k$ at $\theta$ is the same as wavelength of zonal wavenumber $\frac{k}{\sin\theta}$ at the equator. Thus, to get uniform zonal resolution, if one has bandlimit $K$ at the equator, one must apply a low-pass filter to remove frequencies $k > \frac{K}{\sin\theta}$ along each latitude band.

**Hybrid Convolutions.** Conventional CNNs struggle in spherical settings in large part due to the spatial distortion introduced by the curvature. On an equirectangular grid defined in spherical coordinates $\lambda, \theta$, the majority of that distortion comes from the variance in the spacing of $\lambda$ coordinates across colatitudes $\theta$. This limits the use of conventional CNNs to cubed-sphere grids like those used in Weyn et al. (2020). On the other hand, $\theta$ coordinates are uniformly spaced which we can exploit with a factorized hybrid convolution.

Factored spectral convolution was previous employed in Tran et al. (2023). Here, we take the abstraction a step further with a hybrid convolution using a spectral convolution in the zonal direction followed by spatial convolution in the meridional direction. As each latitude has a different radius, we allow each spectral zonal

convolution to learn unique filter coefficients, though we restrict the number of non-zero coefficients per latitude according to the bandlimiting discussed above. We note that using different filters, particularly given that we are not enforcing smoothness constraints across latitudes, does introduce aliasing in the meridional spectrum. However, we do not find this to be an issue in practice.

The largest benefit of the factorized hybrid convolution is efficiency. As the meridional convolution is spatial, we can avoid the full sphere-to-torus DFS transformation by simply using transform-inspired padding along the meridional direction. This circumvents the primary drawback of the full DFS method, which doubles the size of the input data when it appends the glide reflection.

## 5 Experiments

### 5.1 Rotating Nonlinear Shallow Water Equations on the Sphere

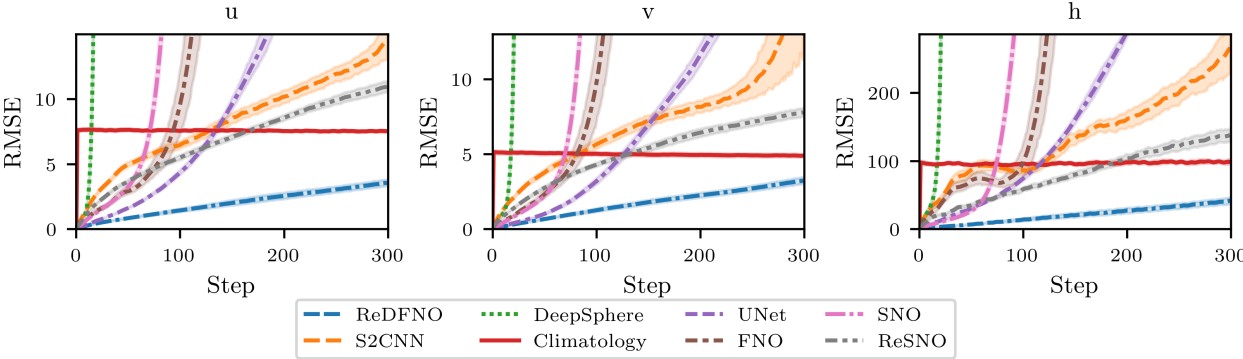

Figure 5: RMSE over time for SWE problem. The ReDFNO demonstrates significantly more stable behavior than both classical spherical architectures and other baselines. While the structural improvements were designed for DFS-based models, they also demonstrate significant stability improvements when applied to spherical harmonic models (ReSNO).

The rotated shallow water equations (SWE) on a sphere are a classic test problem for dynamical cores to be used in large-scale weather and climate models as they capture a number of similar phenomena but are better understood and operate at a more practical scale (Williamson et al., 1992). Here we use the forced hyperviscous equations in two dimensions:

$$\frac{\partial \boldsymbol{u}(\boldsymbol{x}, t)}{\partial t} = -\boldsymbol{u}(\boldsymbol{x}, t) \cdot \nabla_{\boldsymbol{x}} \boldsymbol{u}(\boldsymbol{x}, t) - g\nabla_{\boldsymbol{x}} h(\boldsymbol{x}, t) - \nu\nabla_{\boldsymbol{x}}^4 \boldsymbol{u}(\boldsymbol{x}, t) - 2\Omega \times \boldsymbol{u}(\boldsymbol{x}, t) \tag{9}$$

$$\frac{\partial h(\boldsymbol{x}, t)}{\partial t} = -H\nabla_{\boldsymbol{x}} \cdot \boldsymbol{u}(\boldsymbol{x}, t) - \nabla_{\boldsymbol{x}} \cdot (h(\boldsymbol{x}, t)\boldsymbol{u}(\boldsymbol{x}, t)) - \nu\nabla_{\boldsymbol{x}}^4 h(\boldsymbol{x}, t) + F(\boldsymbol{x}, t) \tag{10}$$

where $\nu$ is the hyperdiffusion coefficient, $\Omega$ is the Coriolis parameter, $u$ is the velocity field, $H$ is the mean height, and $h$ denotes deviation from the mean height. $F$ is a daily/seasonally varying forcing with periods of 24 and 1008 simulation "hour" respectively. The simulations were performed using the spin-weighted spherical harmonic spectral method in Dedalus (Burns et al., 2020) with 500 simulation hours of burn-in where the next three simulation years (3024 hours), were collected for the data set. We use 25 initial conditions for training data and 3 and 2 initial conditions for validation and test data, respectively. Full simulation settings can be found in Appendix C.2.

We use the shallow water equations to examine the impact of our proposed changes to the FNO model and compare our method to more popular spherical deep learning models such as the Spherical CNN (Esteves et al., 2017) and DeepSphere (Defferrard et al., 2020) to illustrate that geometry alone is insufficient to address the observed stability issues. Additionally, we run a comparison against what we call a Spherical Neural Operator both with our modifications ("Restructured SNO", ReSNO) and without them (SNO). The SNO is an FNO implemented using SHT instead of a 2D FFT. As our core model uses the Double Fourier Sphere representation, we refer to it as the Restructured Double Fourier Neural Operator (ReDFNO). We

further include a UNet-structured (Ronneberger et al., 2015) CNN constructed from ConvNext blocks (Liu et al., 2022b) to provide a system-agnostic baseline. All models are implemented in PyTorch (Paszke et al., 2019) and we use TS2Kit (Mitchel et al., 2022) for spherical harmonic transforms. All models were trained for four hours or 40 epochs on 4x NVIDIA A100 GPUs.

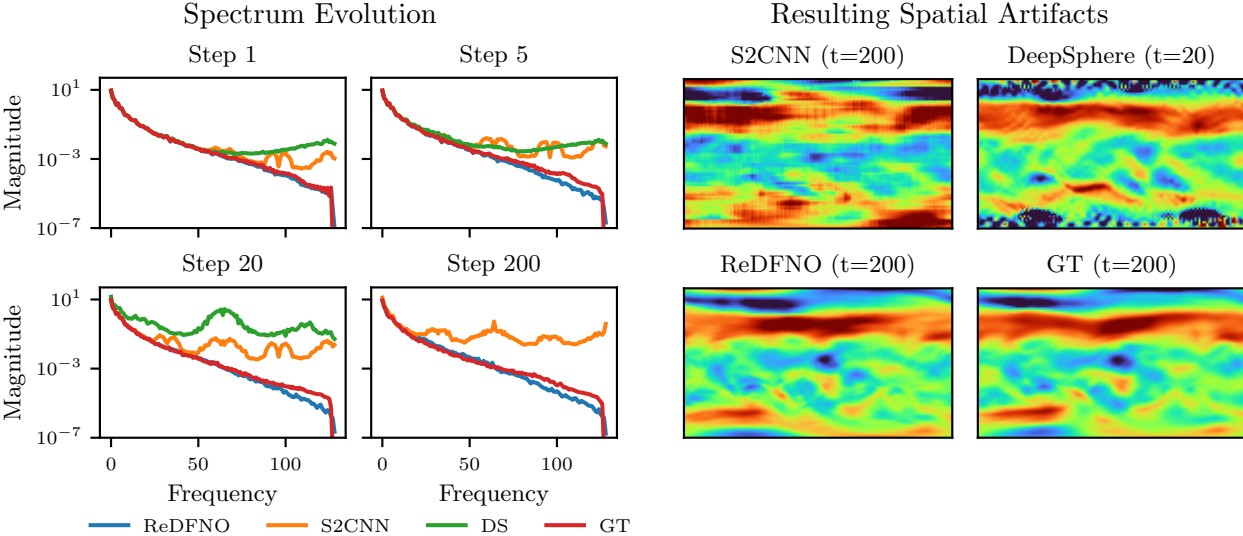

Figure 6: Visualizing failure modes: (left) Both the S2CNN and DeepSphere models have UNet-like structure and so develop spikes in the spectrum corresponding to different resolutions. ReDFNO maintains a consistent spectrum over many timesteps. (right) These spectral artifacts manifest spatially with S2CNN becoming very blocky and DeepSphere developing high amplitude, high frequency patterns near the poles.

**Results.** We present our results in Figure 5. The ReDFNO strongly outperforms all non-stabilized models. The gap between the ReDFNO and ReSNO is striking and can be seen as unexpected given the spherical harmonics do not introduce the same level of distortion as the DFS representation. However, for all spherical harmonic models, TS2KIT uses the Driscoll-Healy (Driscoll & Healy, 1994) SHT. For bandlimit $K$, Driscoll-Healy assumes an input grid of size $2K \times 2K$ while we have a $2K \times 4K$ grid. As a result, the transform truncates half the zonal bandwidth.

Figure 6 is perhaps more informative in the context of the analysis performed in this paper as we can see how the models begin to diverge for architectures with appropriate spherical handling. Both DeepSphere and the S2CNN exhibit similar patterns to Figure 2. Both use UNet-style construction without low-pass filtering on downsampling passes, though the Driscoll-Healy transform used in TS2KIT adds implicit filtering to the S2CNN which could contribute to its more stable behavior. DeepSphere blows up quickly while for the S2CNN, we see spikes develop at the aliases of the zero frequency then grow slowly. This manifests as block artifacts in the full resolution prediction. On the other hand, given our approach is designed around spectral principles, we see minimal spectral artifacts for our stabilized model and in fact, the model can run almost indefinitely without spectral distortion. Over sufficiently long roll-outs, we hit the limit of predictability and the error converges to that of the persistence forecast, as occurs in conventional atmospheric forecasting. We include an ablation of our different model architecture innovations and their effect on long-term stability in Appendix Table 3. Numerical rather than graphical results can be found in Appendix Table 2.

## 5.2 ERA5

In this section, we focus on a real-world application and one of today's grand challenge problems—global weather forecasting. We use the public dataset ERA5 (Hersbach et al., 2020), provided by ECMWF (European Center for Medium-Range Weather Forecasting), which consists of hourly predictions of several crucial atmospheric variables (such as wind velocities and geopotential heights) at a spatial resolution of 0.25° (that corresponds to a $720 \times 1440$ lat-lon grid; or a 25 $km$ spatial resolution) from years 1979 to present day. ERA5

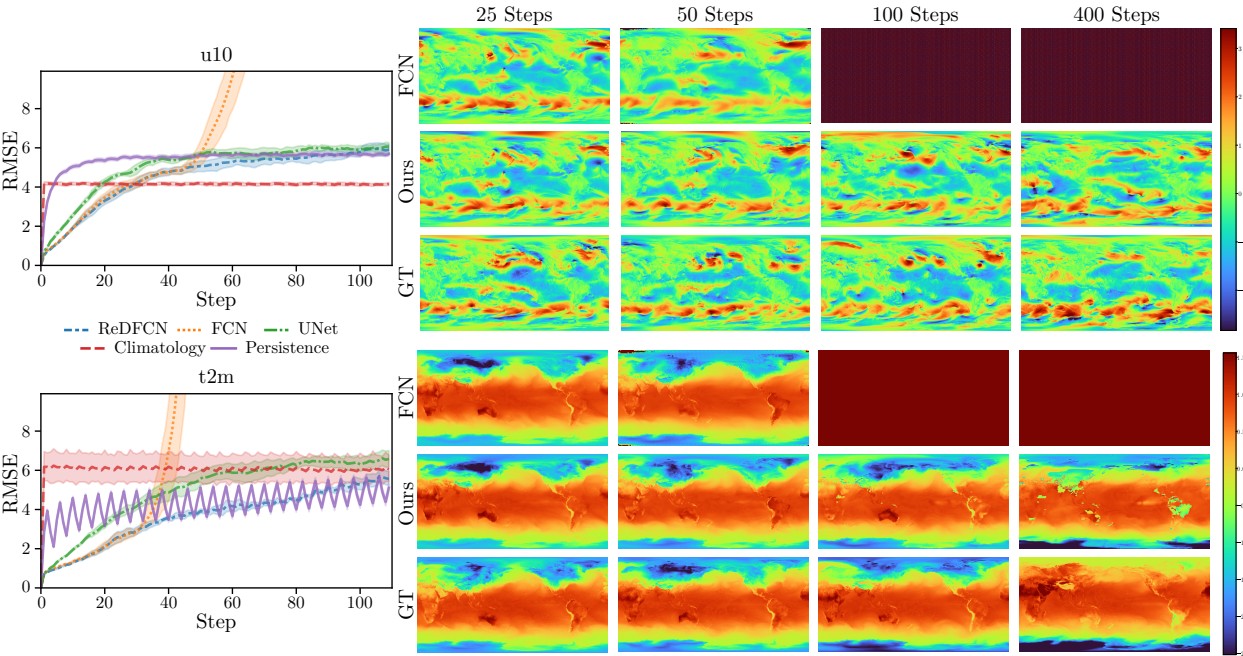

Figure 7: Comparing FCN with our restructured version and ground truth (GT): (left) RMSE by step for *u10m* and *t2m*. (right) Visualizing the forecasts over long time horizons. We can see that while base FCN diverges quickly after its targeted forecast horizon. Our stabilized version produces detailed predictions out to 400 steps (100 days).

is a *reanalysis* dataset that combines observational data with state-of-the-art numerical simulation methods (through Bayesian data assimilation (Kalnay, 2003)). This dataset is widely accepted to be representative of the best available estimate of the atmospheric state of the Earth at any time point. Following FourCast-Net (Pathak et al., 2022), we focus on predicting (forecasting) 20 atmospheric variables at 25 km spatial resolution and 6-hourly time intervals (see Appendix C.3.3 for a list of variables used). Our training and inference pipeline is identical to FourCastNet (and other similar works) with our revised model training for 16 hours across 16 nodes with 4x NVIDIA A100 GPUs per node. Each training data point is a tensor of size $20 \times 720 \times 1440$ (20 channels represent spatial profiles of different atmospheric variables) and the model is trained to predict timestep $t + \delta t$ from initial time point $t$, where $\delta t = 6$ hours. During inference, the model autoregressively predicts further timesteps (similar to SWE).

We apply our model (and input representation) changes to the FourCastNet architecture and demonstrate that our innovations can significantly improve the long-term stability of FourCastNet on several atmospheric variables, which is crucial for sub-seasonal to seasonal time scales (beyond two weeks), and more importantly, for these models to be useful for understanding and analyzing Earth's climate.

**Results.** Figure 7 shows our results on the *u10m* and *t2m* variables. Results for the remaining 18 variables can be found in Appendix C.3.3. The results consistently demonstrate that our changes are able to match the predictive accuracy of FourCastNet while enabling significantly longer prediction horizons. Here, we highlight two fields—for *u10m*, while model error saturates around persistence, the rollout to 400 steps (100 days) shows remarkable level of detail given that the base model shows signs of divergence at around 50 steps. This is an 800% improvement in prediction horizon; for *t2m* field, while our stabilized model does not diverge, we can see one of its lingering weaknesses—we observe unphysical behavior around coasts and mountain ranges (discontinuities) due to Gibb's phenomenon. This behavior appears around step 100 and by step 400 has impacted a significant area. Nonetheless, the model has not diverged at 400 steps.

## 6 Discussion and Conclusion

We showed that we can extend the forecast horizon for autoregressive neural operators significantly at minimal additional cost. For sufficiently smooth fields, like those in the shallow water system and Kolmogorov-forced Navier-Stokes, our design principals allow us to take spectral neural operators that were previously unstable and extend their forecast horizons indefinitely. For more complex and large-scale systems like ERA5, we are still able to demonstrate an 800% improvement in forecast horizon, demonstrating a notable advance towards one of the goals of deep learning-based weather forecast—the ability to extend weather-scale models to seasonal or even climate-length prediction.

**Limitations.** The ERA5 example demonstrates some of the challenges that still exist in fully extending to climate modeling. The imperfect spectral matching in the multi-resolution architecture implies that work remains to fully understand that particular pathology and rectify it. Further, ERA5 has sharp discontinuities along coasts and mountain ranges and we find that with one-step-ahead training, our current methods still fail to learn sufficiently diffusive dynamics to avoid energy accumulation in these areas indefinitely without significant spectral truncation despite significant improvements on those fronts.

**Societal Impact.** Given the difficulty of evaluating long-run behavior, one of the key risks inherent to the development of data-driven models is that their success on short-term metrics could be used by bad actors to discount first-principles projections despite the fact that these models behave unpredictably away from the training manifold. Our work is concerned with reducing that risk by identifying and addressing the issues that cause current models to fail.

While work remains, these results represent a significant improvement in our understanding of autoregressive neural operators for spatiotemporal forecasting. We provided analyses detailing where these models fail and identified principled, low cost architectural changes that can mitigate these failure points. Our experiments reinforced these analyses and demonstrated the success of our innovations, while simultaneously supporting the argument that the failure modes identified apply to more families of nonlinear forecasting models than specifically studied here.

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

# A  Extended Details

## A.1  Pseudospectral Method

Here we perform a step-by-step derivation of the pseudospectral equations from Section 2.1. Recall that we are exploring the nonlinear advection problem for scalar field $u \in \mathcal{L}^2(\mathbb{T})$ defined on the periodic domain $x \in [0, 1]$:

$$\frac{\partial u(x,t)}{\partial t} = -u(x,t)\frac{\partial u(x,t)}{\partial x} \tag{11}$$

Pseudospectral methods compute a solution to the differential equation at a set of $N$ spatial locations $\{x_1, \ldots, x_N\}$ through the use of derivatives computed to high accuracy in the spectral domain (Fornberg, 1996). Assuming that $u$ has bandlimit $K$ where $K = \frac{N}{2}$, we can exactly represent $u$ at the $n^{th}$ collocation point as the discrete Fourier expansion:

$$u_n(t) = \sum_{k=-K}^{K} \hat{u}_k(t) e^{\tilde{k}x_n} \tag{12}$$

where $\tilde{k} = 2\pi i k$ and $\hat{u}_k(t) = \frac{1}{\sqrt{N}}\sum_{n=1}^{N} u_n(t)e^{\tilde{k}x_n}$. In frequency space, computing the exact derivative amounts to elementwise multiplication which by the Convolution Theorem (Smith, 2007) is equivalent to a convolution in the spatial domain, so plugging 12 into 11, we get:

$$\frac{\partial u_n(t)}{\partial t} = -\left(\sum_{\ell=-K}^{K} \hat{u}_\ell(t)e^{\tilde{\ell}x_n}\right)\left(\sum_{m=-K}^{K} \tilde{m}\hat{u}_m(t)e^{\tilde{m}x_n}\right). \tag{13}$$

Expanding the product of summations reveals modes with wave number up to $2K$. In frequency space, we can represent the equation as a system of independent ODEs at each wave number $k = -2K, \ldots, 2K$:

$$\frac{\partial \hat{u}_k(t)}{\partial t} = -\sum_{\ell=-2K}^{2K} (\tilde{k} - \tilde{\ell})\hat{u}_{k-\ell}(t)\hat{u}_\ell(t) \tag{14}$$

$$= \left(\hat{u} * \widehat{\frac{\partial u(x,t)}{\partial x}}\right)[k] \tag{15}$$

which now requires performing convolution in frequency space as well. Naive computation of this convolution is quadratic in the original bandlimit, but this cost can be avoided with fast transform methods like the FFT. Through the dual of the Convolution Theorem, we have that $\hat{u}*\hat{u} = \mathcal{F}(\mathcal{F}^{-1}\hat{u}\cdot\mathcal{F}^{-1}\hat{u})$ and so for pseudospectral methods, we can perform the convolution as elementwise multiplication in the spatial domain with the limiting cost being that of the transform – $O(N \log N)$ in the case of the FFT. Once the time derivatives are computed, one can use any off-the-shelf ODE integrator to march the equation forward via the method of lines (Boyd, 2013).

However, this introduces a complication as the bandlimit of the RHS of Equation 13 is twice that of the original function. For $K \geq \frac{N}{4}$, the bandwidth of the time derivative is now greater than the Nyquist limit. Without increasing the spatial sampling rate, these newly generated modes will alias back onto modes below the Nyquist limit introducing a new source of error.

## A.2 Parameterizing the FNO as a Pseudospectral Method

We begin with the form of an FNO block. Recall that our input is $v^{(\ell)} \in \mathbb{R}^{N \times D^{(\ell)}}$ for layer $\ell \in \{0, \ldots, L\}$ at each discretization point such that $v^{(0)} = u$. $N, K, D$ denote the sizes of the spatial, frequency, and channel dimensions respectively. The block is parameterized by $W^{(\ell)} \in \mathbb{R}^{D^{(\ell+1)} \times D^{(\ell)}}$, and $R^{(\ell)} \in \mathbb{C}^{K \times D^{(\ell+1)} \times D^{(\ell)}}$ and computed as:

$$v_{ic}^{(\ell+1)} = h\left(\sum_{c'=1}^{D^{(\ell)}} W_{cc'}^{(\ell)} v_{ic'}^{(\ell)} + \left[\mathcal{F}^{-1}\mathcal{K}^{FNO}(\mathcal{F}v^{(\ell)})\right]_{ic}\right) \tag{16}$$

$$\mathcal{K}^{FNO}(\mathcal{F}v^{(\ell)})_{kc} = \sum_{c'=1}^{D^{(\ell)}} R_{kcc'}^{(\ell)}[\mathcal{F}v^{(\ell)}]_{kc'} \tag{17}$$

We aim to show how this block can be parameterized in such a way that it is equivalent to:

$$v_{ic}^{(t+1)} = v_{ic}^{(t)} + \eta\frac{\partial v_{ic}^{(t)}}{\partial t} \tag{18}$$

$$\frac{\partial v_{ic}^{(t)}}{\partial t} = v_{ic}^{(t)} \odot [F^{-1}\mathcal{K}^{PS}(\mathcal{F}v^{(t)})]_{ic} \tag{19}$$

$$\mathcal{K}^{PS}(\mathcal{F}v^{(t)})_{kc} = -i2\pi k[\mathcal{F}v^{(t)}]_{kc} \tag{20}$$

where we can assume $\eta = 1$ without loss of generality as it can be absorbed into any multiplicative term. Converting between the two requires one structural adjustment and then explicitly setting our parameters

and activations. For the first step, we move the first term (channel mixing by $W$) outside of the nonlinear activation. We then set $W$ to be the identity matrix giving us:

$$v_{ic}^{(\ell+1)} = v_{ic}^{(\ell)} + h\left(\left[\mathcal{F}^{-1}\mathcal{K}^{FNO}(\mathcal{F}v^{(\ell)})\right]_{ic}\right) \tag{21}$$

Now for $R$, we assume $D_{in} = 1$ and $D_{out} = 2$. The reason for this will become apparent below. We then set $R$ to be:

$$R^*_{kcc'} = \begin{cases} -i2\pi k & c = c' = 1 \\ 1 & c = 2, \ c = 1 \\ 0 & \text{else} \end{cases} \tag{22}$$

otherwise. This gives two channels - the first applies the spectral differentiation operation (20). The second is an identity. We can then define $h$ as a GLU activation (Dauphin et al., 2017; Shazeer, 2020) which splits the channels and performs elementwise multiplication. This completes the mapping by giving us the FNO-like update:

$$v_{ic}^{(\ell+1)} = v_{ic'}^{(\ell)} + GLU\left(\left[\mathcal{F}^{-1}\mathcal{K}^*(\mathcal{F}v^{(\ell)})\right]_{ic}\right) \tag{23}$$

$$\mathcal{K}^*(\mathcal{F}v^{(\ell)})_{kc} = \sum_{c'=1}^{D^{(\ell)}} R^*_{kcc'}[\mathcal{F}v^{(\ell)}]_{kc'} \tag{24}$$

### A.3 Aliasing and Nonlinearity

Aliasing in discretized settings is the result of individual modes being indistinguishable at the given sampling points. For the Fourier basis, we know from the Shannon-Nyquist Sampling Theorem (Smith, 2007) that for a set of $N$ evenly spaced points, we are able to exactly recover signals with bandlimit up to $N/2$ via the Discrete Fourier Transformation (DFT). However, in cases where the signal is not bandlimited, the coefficients returned by the DFT are incorrect as higher modes alias onto lower modes. In this case, if we denote by $U_k$ the true Fourier coefficients that we would obtain in the infinitely sampled regime, the DFT returns coefficients:

$$\hat{u}_k = \sum_{j=-\infty}^{\infty} \sum_{n=0}^{N} e^{i2\pi(k+jN)x_n} u_n \tag{25}$$

$$= U_k + \sum_{j\neq 0} U^{k+jN} \tag{26}$$

If we denote by $U_k$ the true Fourier coefficients one would obtain in the continuous regime, we can observe that aliasing error, the distance between the true truncated function and the function returned by the DFT is defined as:

$$A^2_{|K}[u] = \sum_{k=-K}^{K} (U_k - \hat{u}_k)^2 = \sum_{k=-K}^{K} (\sum_{j\neq 0} U^{k+jN})^2 \tag{27}$$

In many deep learning settings, addressing aliasing has led to improvements in accuracy and generalizability (Vasconcelos et al., 2021; Ribeiro & Schön, 2021; Zhang, 2019; Karras et al., 2021), but these findings are often neglected in the design of new architectures. However, in the autoregressive forecasting setting, these types of issues become a significantly larger obstacle.

In Equation 13, we saw that the nonlinearity in the true nonlinear advection equation results in a bandlimit expansion. This is in fact true of any arbitrary nonlinearity, though the results are less predictable in

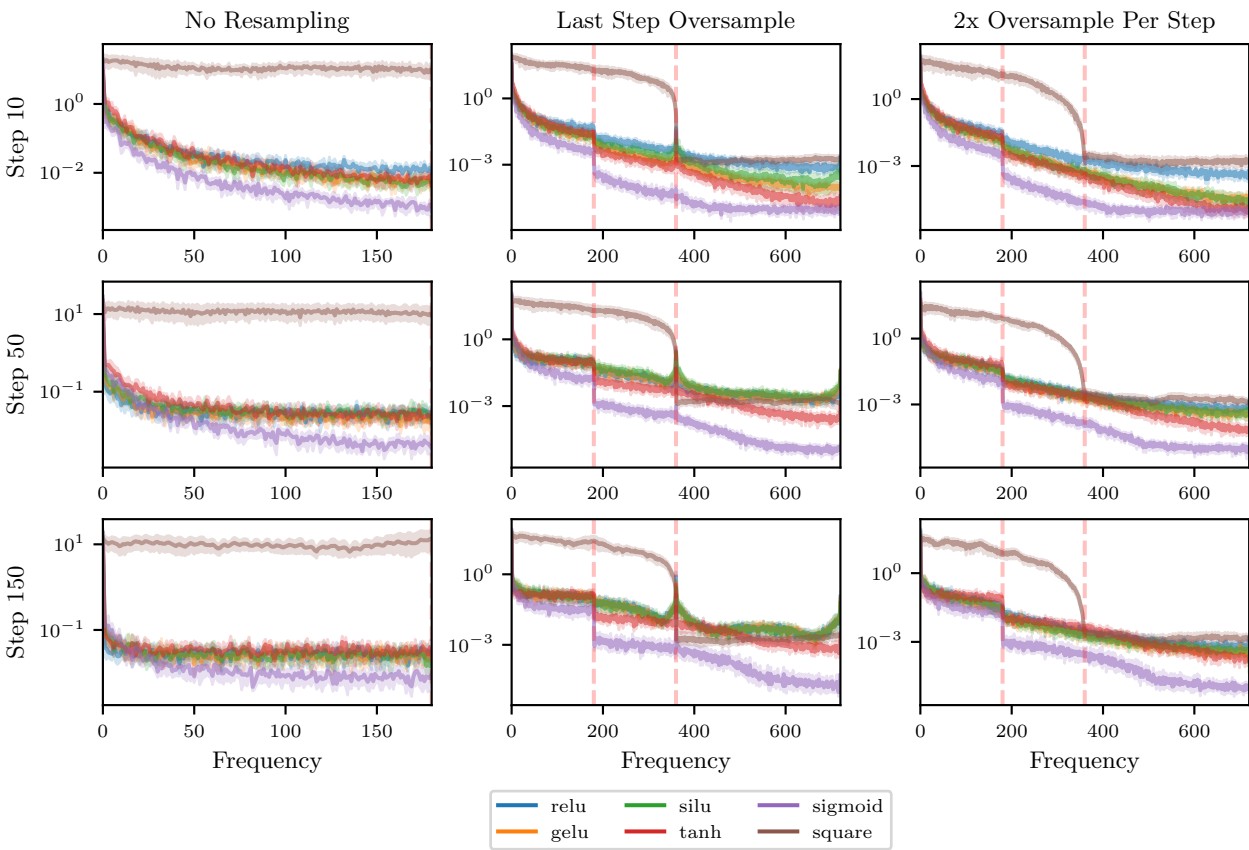

Figure 8: Spectrum evolution of ERA5 data under repeated application of random projections and nonlinearities. Left is at fixed sampling level. Center applies the final nonlinearity at a higher resolution. Right oversamples by 2x before applying every nonlinearity then truncates the top half of the modes to return to the original resolution.

the non-polynomial cases. For continuous nonlinearities like those typically used it deep learning, this can be viewed as an implication of the Weirstrass Approximation Theorem which states that any continuous function on an interval can be arbitrarily closely approximated by a linear combination of polynomials. Exactly characterizing this behavior is in open problem, but we can observe it empirically.

To do this, we perform a simple experiment. We take a sample from one of our experiment datasets, ERA5 and repeat the following procedure:

1. Project via Gaussian random matrix (equivalent to an untrained linear layer) into a larger dimensional subspace.

2. (Optional) Oversample data spatially via Fourier interpolation.

3. Apply nonlinearity.

4. (Optional) Downsample via Fourier truncation.

5. Remix via Gaussian random matrix.

6. Renormalize.

As we can see in Figure 8, the oversampled activation (right) produces drastically different and more consistent spectra compared to the left and center panels. The contrast between the left and center are particularly interesting and relevant for U-shaped architectures. On the left, we see that at fixed resolution, the square function does eventually show signs of a more typical spectral blocking pattern, but repeated application of

the activation functions flattens the spectrum. The right panel indicates this effect is diminished, though it doesn't appear to be eliminated, by explicit anti-aliasing. An interesting aspect is that applying activations to oversampled versions of the flattened spectrums now leads to spikes at the Nyquist frequencies for each resolution. This is a drastic contrast with the anti-aliased spectrum which has no spikes. This experiment shows that while exactly characterizing the impact of aliasing is difficult to analytically model, it does seem to contribute toward the issues we observe in real training scenarios.

## A.4 Comparing Spatial and Spectral Convolution

Spatial and spectral convolution are connected through the Convolution Theorem. Given $N$ uniformly-spaced samples from signals $f, g$, the circular discrete convolution is defined:

$$(f * g)[n] = \sum_{m=0}^{N-1} f[m]g[n-m] \tag{28}$$

Equivalently by the convolution theorem, we could define this as:

$$(f * g)[n] = \mathcal{F}^{-1}(\mathcal{F}f \odot \mathcal{F}g) \tag{29}$$

where $\mathcal{F}$ is the discrete Fourier transform and $\odot$ denotes elementwise multiplication. For large kernels, spectral domain convolution is significantly faster as the limiting factor is the $N \log N$ transform compared to the $N^2$ of the spatial implementation. However, for the small kernel convolutions generally used in spatial CNNs, methods like im2col enable linear scaling for the spatial convolution. In practice using PyTorch, we've found the kernel size where the execution times equalize to be surprisingly small, though in many cases the local response is desirable and spectrally-parameterized convolutions must carry a much larger number of parameters to represent the same filter.

However, in some cases, there are advantages to non-sparse weights as well. If we consider a one-dimensional kernel of size 3, the discrete Fourier transform is computed on the expanded kernel $[w_0, w_1, 0, \ldots, 0, w_{-1}]$:

$$\hat{w}_k = \sum_{n=0}^{N-1} w_n e^{-\frac{i2\pi}{N}kn}. \tag{30}$$

While the typical interpretation of this equation is the inner product of $w$ and the trigonometric polynomial with frequency $k$, we can equivalently view it as the evaluation of the sum of trigonometric polynomials with frequencies $n$ evaluated at $\frac{k}{N}$ weighted by $w_n$. Thus, the resulting frequency response is non-sparse but has a very small number of degrees of freedom as a result of the sparsity of the spatial kernel. In domains where the spectrum rapidly decays, the learning process might then favor using these degrees of freedom to learn patterns relevant to the larger magnitude low frequencies and largely ignore the significantly smaller magnitude high frequencies.

## A.5 DFS transform details

Here we provide additional details of the DFS method (Merilees, 1973; Orszag, 1974). For a field $f$ defined on an equirectangular grid, the DFS representation of $f$ is given by $\tilde{f} : [-\pi, \pi]^2 \to \mathbb{R}$:

$$\tilde{f}(\lambda, \theta) := \begin{cases} g(\lambda + \pi, \theta), & (\lambda, \theta) \in [-\pi, 0] \times [0, \pi] \\ h(\lambda, \theta), & (\lambda, \theta) \in [0, \pi] \times [0, \pi] \\ g(\lambda + \pi, -\theta), & (\lambda, \theta) \in [0, \pi] \times [-\pi, 0] \\ h(\lambda + \pi, -\theta), & (\lambda, \theta) \in [-\pi, 0] \times [-\pi, 0] \end{cases}$$

where $g(\lambda, \theta) = f(\lambda - \pi, \theta)$ and $h(\lambda, \theta) = f(\lambda, \theta)$. The transformed function $\tilde{f}$ is now periodic in both the $\lambda$ and $\theta$ directions. As demonstrated by Orszag (1974), the 2D Fourier basis is equivalent to a complex exponential expansion in the zonal direction and alternating sine/cosine expansions in the meridional direction. Subsequently, much of the recent work in the area focuses on improved basis representations for the DFS

Table 1: Computational costs associated with changes. These are theoretically timings and do not reflect hardware realities and relative costs of different layers in the network. Dimensions are N for the number of points on each spatial axis (assuming H=W=N), K for the kernel size of a spatial convolution, and D for the number of channels in the hidden dimension.

| CHANGE | PARAMETERS | | FLOPs | |
|---|---|---|---|---|
| | OLD | NEW | OLD | NEW |
| DENSE $\to$ DS CONV | $N^2D^2$ | $N^2D + D^2$ | $N^2D^2$ | $N^2(D + D^2)$ |
| DFS $\to$ DFS HYBRID | $2N^2D + D^2$ | $N^2D + D^2 + K$ | $2N^2 \log N + N^2(D + D^2)$ | $N^2 \log N + N^2(D + K + D^2)$ |
| STATIC $\to$ DYN CONV | $N^2D + D^2$ | $2D + N^2D + 3D^2$ | $N^2(D + D^2)$ | $N^2(6D + 3D^2)$ |

method. However, we choose to use the naive function expansion as it simplifies the implementation and allows for more natural handling of vector-valued fields in which we just flip the sign of any field directed across the axis of reflection.

We note the DFS method provides an additional advantage of flexibility in patchified settings like that of the AFNO used in FourCastNet (Pathak et al., 2022). In the patchified setting, the extremal north/south "tokens" are no longer representative of the poles, but rather of large regions surrounding the poles. Efficient spherical harmonic transforms are oblivious to this. As an example, the asymptotically optimal Driscoll-Healy transform (Driscoll & Healy, 1994) employed by TS2KIT (Mitchel et al., 2022), which we use to implement spherical architectures in Section 5, for instance, assigns a quadrature weight of zero to polar samples. This amounts to discarding the information from the entire patch, which for FourCastNet contains 10,080 non-polar samples per variable. This limitation does not exist with the DFS method as we have full control of zonal and meridional bandwidth independently and can adjust bandwidth restrictions to account for the widest region of the patch.

## B  Computational Comparison

While we include timings in Table 1, we include here a table describing the FLOP/parameter comparison on a block-wise basis as the strict timings are affected by memory movement and sequential kernel launches and may not represent an optimized version. In practice, as seen in Table 3, switching to a DS convolution results in a significant speedup despite the slight increase of FLOPs while the Hybrid convolution results a larger slowdown despite the theoretically smaller increase. The practical runtimes could further change as complex support, especially complex mixed-precision support, improves in deep learning libraries.

## C  Experiment Details

### C.1  Navier-Stokes

#### C.1.1  Data

Data is generated using code provided by Tran et al. (2023) to match the settings described by Kochkov et al. (2021). It solves the Navier-Stokes equations (nondimensionalized with Reynolds number $Re = 1000$) for a system with constant Kolmogorov forcing $f = 4\cos(4y)x - .1\mathbf{u}$ across initial conditions. The trajectories are generated using a pseudospectral method at $2048 \times 2048$ resolution and is integrated forward at $\Delta t = .0002$ by a fourth order Carpenter-Kennedy method. 32/4/4 trajectories were generated for train/valid/test with 9764 snapshots per trajectory for a total of 410,088/39,056/39,056 examples.

For the task described in Li et al. (2021b) the sampling occurs at intervals of $\Delta t = 1$ as this large time-step provides a challenge for numerical integration and thus performing well at such a time-step demonstrates a significant speed improvement for the deep learning approach. For our purposes of demonstrating autoregressive growth, we instead sample at $\Delta t = .0035$. The timestep was entirely chosen due to the fact that this

was the interval at which the generation code saved snapshots by default. For consistency with the larger scale tasks, we also predicted $u, v$ values as opposed to vorticity.

### C.1.2 Model Configurations and Training Details

Several architectural choices were made for consistency across models. In each network, appended grid coordinates were replaced by fully learnable position embeddings similar to those used in Pathak et al. (2022). Furthermore, ReLU activations were replaced by the continuously differentiable alternative SiLU (Elfwing et al., 2017). Apart from these changes, each network was generated using the four-layer variant released by the authors of the respective comparison models with hidden dimension of 128.

The ReFNO only uses the modifications from Sections 4.1/4.2 as the domain for this problem is the 2-Torus so modifications to account for spherical geometry are unnecessary. As this is a smaller problem, the depth-separable implementation did not provide enough of a memory savings to increase the hidden dimension as we do in for the Shallow Water example (see: C.2), so we instead augmented the spatial MLP with the bottleneck structure popularized by Sandler et al. (2019) so that run-time was comparable between models.

All models were trained using identical settings. To minimize the impact of hyperparameter tuning, we used the automated learning rate search provided by Defazio & Mishchenko (2023) code for Adan (Xie et al., 2023). During each training step, one input snapshot was provided to the model for the purposes of predicting the output at $t = t_0 + \Delta t$. These were optimized using mean-squared error rescaled by the mean norm of the dataset to reduce the impact of precision issues for 20 epochs at batch size of 128 per run. Error bars are the 95% confidence intervals produced by sampling over 7 individual training runs per model evaluated across all initial conditions in the test set.

## C.2 Rotated Shallow Water

### C.2.1 Data

We generate the data from the hyperviscous, forced shallow water equations:

$$\frac{\partial \boldsymbol{u}(\boldsymbol{x}, t)}{\partial t} = -\boldsymbol{u}(\boldsymbol{x}, t) \cdot \nabla_{\boldsymbol{x}} \boldsymbol{u}(\boldsymbol{x}, t) - g \nabla_{\boldsymbol{x}} h(\boldsymbol{x}, t) - \nu \nabla_{\boldsymbol{x}}^4 \boldsymbol{u}(\boldsymbol{x}, t) - 2\Omega \times \boldsymbol{u}(\boldsymbol{x}, t) \tag{31}$$

$$\frac{\partial h}{\partial t} = -H \nabla_{\boldsymbol{x}} \cdot \boldsymbol{u}(\boldsymbol{x}, t) - \nabla_{\boldsymbol{x}} \cdot (h\boldsymbol{u}) - \nu \nabla_{\boldsymbol{x}}^4 h + F \tag{32}$$

where $\nu$ is the hyperdiffusion coefficient, $\Omega$ is the Coriolis parameter, $\boldsymbol{u}$ is the velocity fields, $H$ is the mean height, and $h$ denotes deviation from the mean height.

The forcing F is defined:

```python
def find_center(t):
    time_of_day = t / day
    time_of_year = t / year
    max_declination = .4 # Truncated from estimate of earth's solar decline
    lon_center = time_of_day*2*np.pi # Rescale sin to 0-1 then scale to np.pi
    lat_center = np.sin(time_of_year*2*np.pi)*max_declination
    lon_anti = np.pi + lon_center   #2*np.((np.sin(-time_of_day*2*np.pi)+1) / 2)*pi
    return lon_center, lat_center, lon_anti, lat_center

def season_day_forcing(phi, theta, t, h_f0):
    phi_c, theta_c, phi_a, theta_a = find_center(t)
    sigma = np.pi/2
    coefficients = np.cos(phi - phi_c) * np.exp(-(theta-theta_c)**2 / sigma**2)
    forcing = h_f0 * coefficients
    return forcing
```

This is not designed to mimic an exact physical process, but rather to force some level of daily/annual pattern. It consists of two Gaussian blobs centered on opposite sides of the planet that circle on a daily basis. Each Gaussian increases/decreases $h$ with the intensity of the forcing increasing towards the center. The axis of rotation for these Gaussians varies over a model year which we define to be 1008 model hours.

Integration is performed forward in time using a semi-implicit RK2 integrator. Step-sizes are computed using the CFL-checker in Dedalus. The 3/2 rule is used for de-aliasing. Background orography is taken from earth orography and passed through mean-pooling three times (until the simulations became stable empirically). Hyperdiffusion is matched at $\ell = 96$.

Initial conditions are randomly sampled from ERA5. $u, v, z$ are taken from the hpa 500 level with $z$ used as $h$ is the shallow water set-up. We found that these conditions did not produce stable rollouts at the given hyperdiffusion level without pre-filtering the data, so prefiltering was performed by executing ten iterations of 50 steps followed by solving a balance BVP. This was likely more than necessary, but given the quantity of data generated, we wanted to avoid in-depth manual inspection of the data.

As the diffusive system is decaying over time, we initialized each run with 500 burn-in hours then took the next 3 model years of data for a total of 3024 samples per year. In total, the training set consisted of 25 trajectories of length 3024 for a total of 75,600 samples. Validation and test used an additional 2 and 3 trajectories respectively.

### C.2.2 ReDFNO Architecture

The ReDFNO utilized four blocks structured as described in Figure 3. Each block is defined with embedding dimension 192. Our frequency-domain spectral normalization is used for spectral layers with conventional spectral normalization is used for spatial layers. Hybrid convolutions with DFS padding are used for the spectral block. We found a slight boost to stability by using a nonlinear spectral block as used in the AFNO. This structure is described in Figure 3. In total, this network contains approximately 17 million parameters.

We utilized time-dependent position embeddings. Position embeddings have previously been used for neural operators in both the AFNO (Guibas et al., 2021) and FCN. However, in cases where external forcing data is not directly available, static position embeddings are unable to fully represent seasonal behavior in earth systems. To address this, we introduce dynamic, time-dependent position embeddings. These embeddings are generated through the use of a shallow MLP that combines static embeddings with either sinusoidal features representing daily and yearly periods or more informative features directly computed for a given problem like solar declination.

### C.2.3 Comparison Configurations

As our model used 17 million parameters, we tuned and augmented comparison models until the sizes were roughly comparable when scaling laws allowed it. Otherwise for FNO based models, we used as similar settings as possible, though the vastly different parameter scaling led to higher parameter counts. We list model dimensions in Table 2. The S2CNN model uses filters of [20, 16, 12, 8] for interpolates, and the DeepSphere uses kernel size 5 rather than the default 3, as we found these to achieve better results. Our UNet baseline uses ConvNext (Liu et al., 2022b) blocks with three blocks per stage instead of the classical blocks from Ronneberger et al. (2015) as we felt modernizing the architecture was important for fair comparison. We found this worked considerably better in tests.

Note that persistence and climatology are two constant forecasts. Persistence uses the initial condition as the forecast while climatology implies using the mean estimate over the dataset.

## C.3 ERA5

### C.3.1 ReADFNO Architecture

This network uses FourCastNet as a base. The core processing unit consists of 8 AFNO blocks using our factorized nonlinear spectral convolutions rather than the standard complex MLP. As the AFNO architecture

Table 2: Model dimensions for SWE experiments, along with the 1-step mean absolute error (MAE) and 100-step MAE. All MAE are reported in units of $10^{-2}$.

| MODEL | STAGES | HIDDEN DIMENSIONS | #PARAMS | 1-STEP MAE | AVG. 100-STEP MAE |
|---|---|---|---|---|---|
| UNET | 4 | [64, 128, 256, 512] | 16M | .57 | 31.4 |
| FNO | 4 | 64 | 536M | 0.80 | 19.2 |
| SNO | 4 | 64 | 268M | 0.65 | 26.6 |
| RESNO | 4 | 192 | 9M | 1.18 | 22.6 |
| **ReDFNO** | 4 | 192 | 17M | **0.49** | **4.1** |
| S2CNN | 4 | [64, 128, 256,512] | 18M | 1.63 | 36.0 |
| DEEPSPHERE | 5 | [64, 128, 256, 512, 1024] | 38M | 1.74 | NAN |

Table 3: Ablation path for SWE experiments evaluated on validation set. In descending order, each row adds new feature on top of previous inclusions until we reach the full stabilized architecture. Step is one training step (forward and backward) at batch size 16 on a NVIDIA A100 GPU. (DS=depth separable convolution, DFS=Double Fourier Sphere transform, SN=Spectral Normalization, Path=Reorder+Nonlinear, Shaping=Latitude-wise spectral truncation)

| MODEL | #PARAMS | STEP (S) | MSE (T=1) | MSE (T=20) | MSE (T=40) | MSE (T=80) |
|---|---|---|---|---|---|---|
| FNO | 536M | .20 | 8.1E-5 | 2.6E-4 | 1.0E-2 | INF |
| +DS (x3 WIDTH) | 25M | .22 | 1.2E-4 | 1.0E-2 | INF | INF |
| +DFS | 27M | .40 | 8.2E-5 | INF | INF | INF |
| +SN | 27M | .46 | 8.7E-5 | 4.5E-4 | 8.0E-4 | 1.4E-3 |
| +PATH | 27M | .46 | 9.5E-5 | 3.6E-4 | 5.6E-4 | 1.0E-3 |
| +SHAPING (**ReDFNO**) | **17M** | .41 | **8.0e-5** | **2.3e-4** | **3.7e-4** | **6.3e-4** |

is more complication and non-isotropic, we made several further changes to eliminate aliasing-inducing operations:

1. Downsampling - Strided convolutions perform decimation which is addition in the spectral domain. To avoid this, we downsample via Fourier interpolation/truncation in the DFS representation. During downsampling, channels are expanded via 3x3 spatial convolution. This occurs at $1x/2x/4x/8x$ downsampling levels with the DeADFNO operations occuring at $8x$ downsampling as in FourCastNet.

2. Upsampling - We introduce a Unet-like structure here. At each intermediate resolution, the a downsampled version of the original data is added back to the upsampling path then fed through a single layer spatial CNN. The spatial CNN is used here as the spectral convolutions become increasingly memory-hungry as resolution increases so we can make best use of them at the lowest resolution.

3. LayerNorm $\rightarrow$ InstanceNorm - LayerNorm divides each pixel by a spatially varying standard deviation estimate. While we've observed the impact to be minor, this does produce new frequencies. We therefore replace it with InstanceNorm which computes the standard deviations over the spatial dimensions of a channel rather than the channel dimensions of a particular pixel.

The model was trained using the Nvidia Apex implementation of LAMB (You et al., 2020) following the schedule of FourCastNet. We found that the gradient rescaling in LAMB acted similarly to a trust region and avoided training instability we experienced with Adam in this case. However, we were able to obtain similar partial training performance using a version of Adam with added step size constraints, but chose to stick with the known optimizer for this work.

### C.3.2 Comparison Configurations

FourCastNet comparisons use the pre-trained model weights provided by the paper authors. Similar to SWE, the UNet comparison uses ConvNext blocks rather than conventional ResNet structure. The UNet was trained using the Adam Kingma & Ba (2017) optimizer with learning rate selected by grid search over $[1e-5, 5e-5, \mathbf{1e-4}, 5e-4, 1e-3]$. Epochs and schedule follow settings in FourCastNet.

The smaller number of comparisons here is due to the fact that the majority of the architectures used in the SWE experiments are incapable of scaling to the larger input sizes without seriously diminishing network capacity.

### C.3.3 Additional Results

Full field by field RMSE can be found in Figure 9. Convergence to persistence was obtained for all fields except for Surface Pressure and sometimes Z50. In the FourCastNet training recipe, all fields (normalized) are weighted evenly. Surface Pressure has relatively low snapshot-to-snapshot variation but has significant outliers as it tends to be proportional to altitude. In the normalized-but-not-weighted scheme, this tends to lead to errors in this field being undervalued. Z50, on the other hand, is a slowly varying smooth field, but is quite distant ( 15KM) from the next closest pressure level recorded in this ERA5 subset. We suspect this underperforms for this reason.

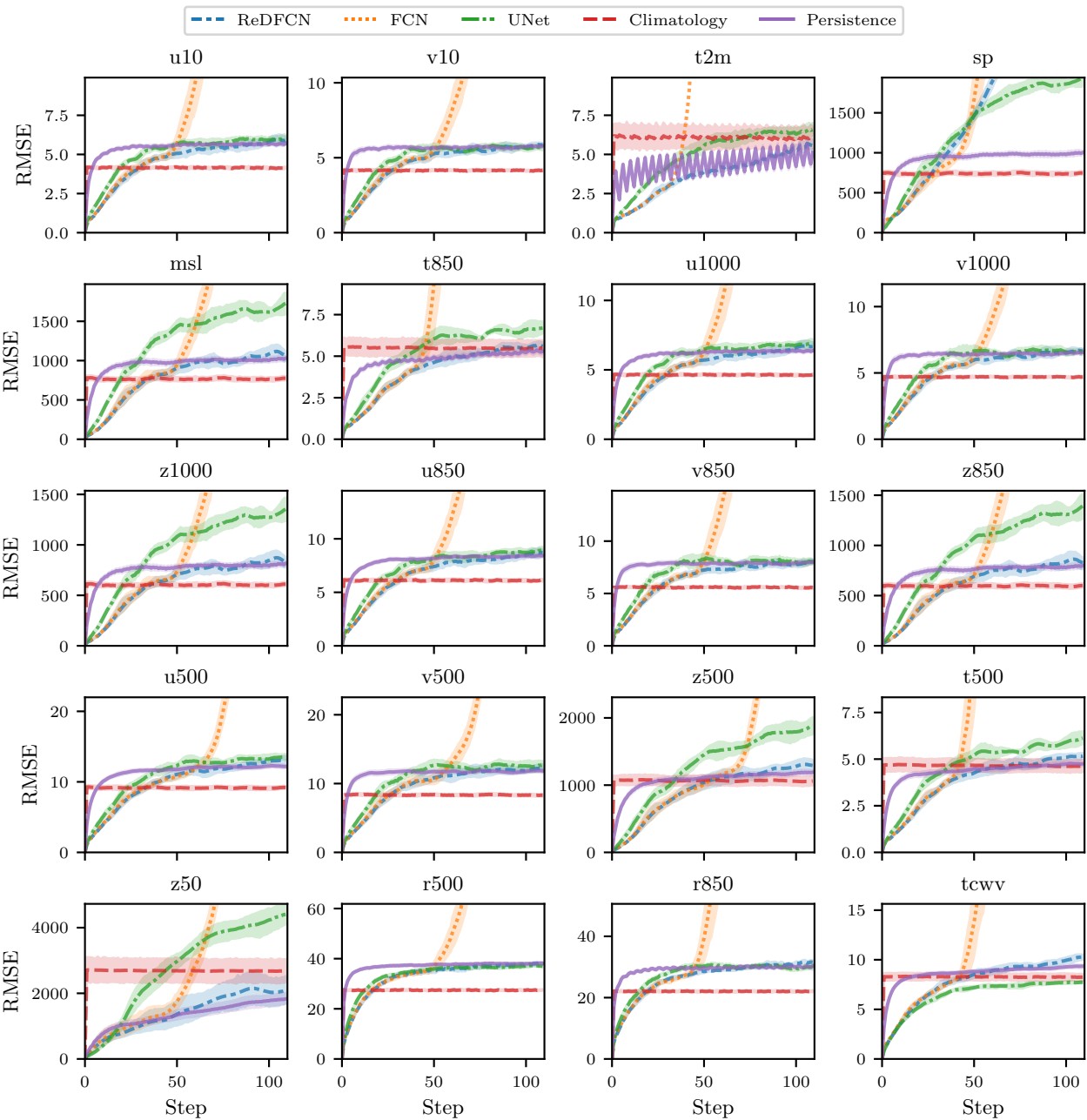

Figure 9: The restructured model matches the performance of the two-step-ahead trained FCN, but demonstrates improved stability and asymptotically converges to the persistence forecast performance for all but the surface pressure field.

