# OpenReview forum: "Towards Stability of Autoregressive Neural Operators"
_TMLR — Accepted by TMLR_

### Review · Reviewer_Hdii · 2023-07-03

**Summary Of Contributions:**

The paper introduces a methodology for developing neural autoregressive models that can make long-term forecasts without accumulating integration error resulting in a total collapse. The paper takes modeling spatiotemporal dynamics, in particular Earth climate dynamics, as a use case. It reports strong results on the ERA5 data set, that is often treated as the most advanced climate model used by climate researchers.

**Audience:**

No

**Broader Impact Concerns:**

The paper addresses the broad impact concerns in Section 2 under the “societal impact” paragraph. However, the provided discussion is from the perspective of the ethical risks associated with the studied problem setup and not about the particular solution the paper presents.


**Claims And Evidence:**

Yes

**Requested Changes:**

The following changes would improve the paper:

 - Provide a more detailed and more accessible background section where all terms not familiar to the machine learning research readership are properly and simply explained. See a nonexhaustive list of these terms under the “weaknesses” paragraph above.

 - Discuss the generalization aspects of the proposed methodology, if possible illustrated on a small additional data set.

 - Clarify the relationship of PDE modeling to the rest of the material more clearly.

**Strengths And Weaknesses:**

**Strengths:**

The paper studies an advanced topic with immense societal impact. Its proposed methodology sketched out in Figure 3 is simple, intuitive, and novel. The results reported on the ERA5 data set are intriguing. The 100 and 400 step results show a clear improvement over FCN. This may be a significant outcome for climate researchers. Figure 1 demonstrates the limitations of FCN very clearly, providing solid motivation for the proposed solution. Likewise, the illustration of the aliases in Figure 2 is also helpful to the reader. The paper provides a comprehensive survey of the related literature and positions its contribution properly.

**Weaknesses:**

I am not confident that the current shape of the paper speaks directly to the machine learning audience. Although it uses a good deal of machine learning tools, it builds on many concepts outside the scope of a typical machine learning researcher. The paper either assumes these terms to be known or gives too brief an explanation of them for an outsider to climate research. Example terms with this character: collocation point, advection equation, spectral differentiation matrix, pseudospectral methods, double Fourier sphere method, geometry-aware filter. It is also not clear in the storyline where exactly PDEs come to play and how they are related to the workflows given in Figure 3. The paper would also benefit from a discussion about the generalizeability of its findings to other modalities than climate data, and if possible also beyond spatiotemporal dynamics modeling.

---

> ### Author Response · Authors · 2023-08-21
>
> We thank the reviewer for their detailed reading of our work and for their perspective. We've updated our submission in response to the feedback we've received. Before going into the line-by-line responses, we feel it is important to address the issues that the reviewer has brought up concerning whether or not this is the correct audience for such a paper. We include a longer standalone response in a global official comment, but we want to address it briefly here as well.
>
> Machine learning for the physical sciences is an area that TMLR has previously considered a relevant audience. In the month prior to our submission, the journal accepted three separate papers closely aligned with our own submission [1][2][3], two of which were specifically FNO improvements [1] or connected to spectral methods [2]. Top tier conferences such as NeurIPS and ICML have prominently featured this subject area as well through workshops and submission categories.
>
> We feel very strongly that this is an appropriate venue for this work and that bringing together insights from numerical methods and machine learning provides value to machine learning researchers working on problems in the computational sciences. We appreciate your perspective on areas where this connection can be bridged more cleanly which we aim to address in both our updates and in the line-by-line responses:
>
> __Requested Change 1__
>
> We appreciate the feedback here. You've given us a great deal of information that will help us improve the clarity of our work. Many of these terms are currently defined in the text, but we’d like to make this clearer:
> - Since the term “collocation” may confuse unfamiliar readers more than it helps readers who are already familiar with pseudospectral methods, we can drop this.
> - Advection equation - The nonlinear advection equation is exactly equation 1 which is defined in the next line. We can update this to “the following nonlinear advection equation” to make it clear that it is describing the next line.
> - Spectral differentiation matrix - Thanks for pointing this out. We’ve moved this discussion to the appendix where it can be described more thoroughly and now only highlight the most important connection in this section.
> - Pseudospectral methods - Section 2.1 walks through the development of a basic pseudospectral method. The term itself is described in the second paragraph of 2.1 by differentiating it from the spectral methods defined earlier in the paragraph while the paragraphs from equation 1 and down walk through an extended example of the pseudospectral method in the Fourier domain. Additional supporting derivations are included in the appendix.
> - DFS/Geometry-aware filter - The term DFS is currently used in the heading of 3.2 - this was an accident resulting from an earlier re-organization and we will remove it as the term has not been defined at that point in the text. Apart from that, as these are parts of our contributions, when these terms are initially used in the introduction, it is not expected that the reader knows what they are yet. We describe these in Section 4.3 when we walk through our approaches for mitigating the issues introduced in Section 3.2.
>
> __Requested Change 2__
>
> We agree that demonstrating performance across multiple problems is important. We demonstrate performance on three different datasets in the paper. The first two - simulations derived from the Navier-Stokes and the Shallow Water Equations are both general fluid transport problems. Fluid transport is a core component of weather and climate, but it is a much broader field than that and is often used as the “real world” example in many papers on neural operators.
>
> While prior work that we reference in the related work section explores the impact of aliasing in other settings where it has been found to hurt generalization and reduce equivariant behavior, the error accumulation issue we explore here is specific to autoregressive spatiotemporal modeling, so we would not expect to see this occur in other settings.
>
> __Requested Change 3__
>
> The connection between our work and PDEs is a matter of problem setting. Our interest is in spatiotemporal fields generated by physical phenomena, which can often be described by PDEs. The traditional numerical methods used to solve these equations are valuable for analysis in the sense that they show us which representations and computational approaches can efficiently represent these fields. Trajectories produced by PDE solvers become valuable test cases for data-driven methods. However, data-driven methods like the FNO are not PDE solvers. They are fully data-driven and do not require knowledge of the underlying equations. This means that they can be used generally for real-world spatiotemporal physical data as well as simulation data, as we demonstrate on the ERA5 dataset.
>
> [1] https://openreview.net/pdf?id=EPPqt3uERT
> [2] https://openreview.net/pdf?id=wNBARGxoJn
> [3] https://openreview.net/pdf?id=j3oQF9coJd’

---

### Review · Reviewer_gvhB · 2023-07-20

**Summary Of Contributions:**

The authors investigate various techniques to improve the long-range forecast stability of Fourier Neural Operator techniques. They rely on insights from spectral and pseudospectral methods to identify architectural augmentations that lead to more accurate roll-outs of their techniques. The study is very well put together, has significant potential for impact, and I support its publication.

**Audience:**

Yes

**Claims And Evidence:**

Yes

**Requested Changes:**

See comments.

**Strengths And Weaknesses:**

Strengths:

1. The key strength of this article is that it attempts to address the issue of unstable and unphysical behavior of geophysical surrogates during long-term rollouts. These limit the accuracy and computational gains of surrogates for practical applications and fall short of certain important challenges in the atmospheric forecasting community - subseasonal forecasting. I encourage the authors to continue their research in this direction. I also think the idea of adaptively learning filter transfer functions is very interesting.

2. The connections drawn by the authors to well-known ideas in spectral methods and numerical analyses are timely and connect two rich fields. That being said, identifying aliasing errors and actually handling them appropriately are two different things and the techniques proposed by the authors are quite valuable for surrogate modeling of dynamical systems. Connections to computational complexity for different techniques are also appreciated.

3. The experiments, appendix, and code provided by the authors are extensive and thorough and I commend them for encouraging the reproduction of their results.



I do not have specific weaknesses but list out a few comments/questions below:

1. Why does the UNet remain stable for the experiments in section 5.2 but diverge in Section 5.1? Have the authors looked into this? I'd be curious to learn why this is the case.

2. Have the authors looked into how the filter transfer functions for their NL Spectral convolution evolve over time? Is there any connection to the typical frequencies expected in their physical experiments.

3. What is the training and memory overhead (or benefit) of the authors proposed additions to the standard FNO? Is this a-priori quantifiable for a given problem?

4. Can the authors comment on how a pretrained model with stability issues may be fixed in a post-hoc manner using some insights from this research?

---

> ### Author Response · Authors · 2023-08-21
>
> We thank the reviewer for the thorough reading of our work! We agree that many of these questions are highly interesting and we will do our best to answer them.
>
> 1. This is mostly a function of the cutoff times of each graph. In Figure 2, you can see that while the UNet has not diverged, the high frequency components of the UNet are becoming increasingly oscillatory as time progresses. By step 100, while this is not included in the text, the UNet shows significant unphysical patterns near the poles and if we roll out a bit longer, this does end up diverging as well. This lines up with Figure 5 from the SWE experiments where the UNet is also a late diverger.
>
>     That said, while this work focused on FNO-type architectures, in a speculative capacity, we do suspect that it is possible to adapt most of our proposed changes to spatial convolutional architectures given the underlying operations are virtually identical - the joint spectral-spatial nature of the FNO just make the design and analysis cleaner so we restricted the scope to that setting in this work.
>
> 2. We’ve looked into this, but found that on such high dimensional feature maps, it can be hard to tie what is observed in individual layers to the final result, so we elected to report analysis on model outputs only. We believe applying interpretability techniques in this setting could be an interesting direction for future work.
>
> 3. Thanks for this suggestion. We agree it would be useful to include and have added an appendix table describing this. One caveat though is that while these are exactly quantifiable in theory for parameters and FLOPs, we’ve found that implementation considerations like memory movement and complex number support  (especially mixed-precision support ) tend to have a larger impact on actual performance than the exact numbers. The new table depicts the theoretical numbers, but the ablation table in B1 (Now C1) gives a better picture of actual performance differences, at least as of PyTorch 1.10. As a side note, our strategy was generally to take whatever capacity we saved from one change (DS Conv, for instance) and apply that capacity in other places (increased width and dynamic filters), so the overall cost of the network did increase.
>
> 4. This is an area we performed fairly extensive experiments on, though the results were inconclusive and as such we did not include them in this work. For instance, the most successful intervention for post-processing was probably low pass filtering. This, however, came with an accuracy trade-off since a model trained to use high frequency information often doesn’t perform as well on short-horizon forecasts when that is not available. On longer time horizons, we generally found improved stability, but there was often also a decrease of norm overtime which eventually resulted in fields dissipating to zero. It might be possible to address this with some type of global norm correction as a counterbalance, but we opted to explore ways to bias the model towards learning filtering behavior during the training process rather than explicitly applying fixed filters. It is an interesting area for future work, but we feel would be a digression in the current submission.

---

### Review · Reviewer_Cgrj · 2023-08-13

**Summary Of Contributions:**

This paper proposes a new network architecture for autoregressive modelling of spatiotemporal physical systems described by partial differential equations. The authors first analysed aliasing issues in popular neural architectures, such as the Fourier Neural Operator, through the lens of pseudospectra methods. A key insight is that convolutions in frequency domains, a popular operation in these models, introduce
aliasing. The authors then proposed a few architectural changes and other application-specific designs. The experiments on simple synthetic and larges-scale real datasets show improvements of their methods compared to other methods.

**Audience:**

Yes

**Broader Impact Concerns:**

The ethical implications are discussed, and I think this is adequate.

**Claims And Evidence:**

No

**Requested Changes:**

Critical: please include these changes or address the questions.
1. Please clearly establish the connection between the pseudospectral methods and the FNO in Eqns 3,4. Are these equations implementing 1x1 convolution in temporal and spectral spaces, mentioned later?
2. The authors try to explain how FNO maps to Euler's method. Please write down how each part of (3) maps to the Euler, in equations.
3. Write an equation for what is happening in **Depthwise Separable Spectral Convolution**. Which part of (3) maps to the separable spectral convolution, if they are related?
4. Explain how spectral normalization can be implemented with polar form. I'm sure this is simple enough for the authors to explain with equations.
5. Explain clearly what's done in the methods in Section 4.2. I have zero idea about what's happening here.
6. Section 4.3 is better than 4.2, but I am still a little confused. Are the two bolded methods addressing the same or related issue caused by polar coordinates causing different spatial frequencies?

Others:
1. The last sentence of the top paragraph on page 6 mentions distributional shift, but how is this dealt with? Did I miss this somewhere?
2. It would be very useful to include arguments of functions in all differential equations (1,2,6,7)
3. The term "residual" is mentioned in FNO context and also the residual network context. The former needs clarification.

**Strengths And Weaknesses:**

## Strengths
1. The paper addresses the growing research field of ML for scientific applications, particularly weather prediction.
2. Overall, The method looks sound and reasonable, and the improvements are substantial in some aspects of the applications tested.
2. The experiments are thorough and adequate, the authors compare with a few baselines and performed ablation studies, clearly showing the contributions of each modification from FNO

## Weaknesses
1. The paper is very hard to follow. I really struggled to get a faint picture of the paper, and I am sure I will be missing technical details unless the authors submit a much-improved revision. This is probably due to my background being in machine learning rather than numerical methods or signal processing, so I might be missing some details, but for an ML journal and audience, I personally think the writing needs to be significantly improved with sufficient details presented more clearly. I feel like many important parts of the paper, such as Sec 4.2, are personal notes of the authors that are only accessible to experts. The paper has to be enriched with equations and clearly defined variables.
2. If I understand correctly, the problem of aliasing arises during spectral convolution, and how this is related to Equations 3 & 4 of FNO is not clear at all. Is the matrix multiplication in Equation 4 the convolution?
3. The claim that the model can achieve 800% improvement in stable prediction range feels like a overclaim. This is established on the results of a single channel out of 20 in the data. The prediction of other channels compared to the baselines need to highlighted, unless the field generally agrees to look at the best-predicted channel.

---

> ### Author Response · Authors · 2023-08-21
>
> We greatly appreciate the time and energy spent by the reviewer on providing such useful and insightful feedback. We've updated the submission with a number of modifications we made as a result of the review and hope the clarity is improved. We’ve split this into two chunks: one addressing the weaknesses and one addressing the requested changes:
>
> ## __Weaknesses__
>
> __W1__
>
>  We’ve taken many of the reviewer's suggestions and included the explicit equations where the reviewer noted the current submission was unclear. We've also updated the wording on areas that seemed to be points of confusion across reviews (the source of aliasing, for instance). Other sections like 4.2, we have completely rewritten.
>
> __W2__
>
> Aliasing occurs as a result of nonlinearities applied in the spatial domain. In the current text, this is described through the example in 2.1. We agree it should be more clear which specific parts of Eqs. 3 and 4 introduce aliasing, so we have clarified that there and in most sections discussing aliasing.
>
>   The issue in equation 3 is actually that the spectral convolution (the second term, which is broken out into EQ 4) is the only term capable of mitigating aliasing by filtering high frequency modes, but the pointwise linear term skips the convolution while also passing through the nonlinearity which makes it very difficult for the model to learn to deal with aliasing. The updated version of Section 4.2 where we discuss aliasing mitigation explains this in more detail.
>
> __W3__
>
> Thank you for raising this concern. We include only two fields in the main body because we were aiming to fit into the 12 page standard size for TMLR and these figures are large. We still feel that including them in the main body would hurt readability, but we’d be happy to add the rest in the appendix (~10 pages). We do include plots of RMSE for each channel in the appendix, and observe substantial stability improvements across the board. We'd also be happy to remove the specific numerical claim as we do feel the results stand for themselves.
>
>  This more limited claim is specific to the hardest task we evaluate on. In the shallow water equation experiments, which is a challenging set-up in its own right, we have not seen divergence from our model at any point in autoregressive inference. The plot we include in 5.1 ends at 300 steps because we wanted the differences between the comparisons to be visible, but we could add a longer plot to the appendix demonstrating even longer time horizons if that would make the stability improvements clearer.
>
> (Continued)

---

> > ### Author Response · Authors · 2023-08-21
> >
> > (Continued from above)
> >
> > ## __Requested Changes:__
> > __RC1/RC2__
> >
> > Thank you for raising these issues. It is valuable for us to know where we can improve clarity. We’ve significantly updated this section so that it focuses on the connections that are most relevant to understanding our work and adding the full derivation to the appendix for interested readers.
> >
> > To further improve clarity, we’ve updated the notation used in 2.2 to show explicit summation over specific axes so that readers should no longer need to infer which operation is performed by examining the shapes of the relevant tensors and also specify which phrases we use to describe which components of the equation. The updated notation should also make the difference between Eq 4 and the newly added depthwise separable spectral convolution equation in 4.1 clearer.
> >
> > __RC3__
> >
> > Thank you. This is a really helpful suggestion. Depthwise separable spectral convolution is a lightweight variant of spectral convolution which is described in (4), but we agree that it would be beneficial to write out these equations and have now done so.
> >
> > __RC4__
> >
> > The last line of this paragraph is actually exactly the equation. To make this clearer, we’ve aligned the notation with the new equation we’ve added in response to item 3 and broken the precise definition out into its own line so that it stands out more strongly.
> >
> > __RC5__
> >
> > We agree the original Sec. 4.2 was too brief, and as this is one of our important contributions we have re-written it. In the new version we explicitly motivate the choices made in re-arranging the FNO block to explain how we arrived at the design shown in Figure 3.
> >
> > On 4.3, The Double Fourier Sphere transform is meant to address the artificial discontinuity assumed by the FNO (and related methods) at the poles. However, a naive implementation of the DFS method has two major disadvantages over using something like a Spherical Harmonic transform (though the transform itself is significantly cheaper) which is what we seek to mitigate in the bolded sections. We’ve now explicitly stated this in the text.
> >
> > The first of these issues is geometric distortion. DFS treats all latitude lines as if they have the same circumference. The first bolded section, “Shaping with Geometry-aware Filters” mitigates this issue by enforcing uniform resolution over the sphere through filtering. The second issue is that DFS functionally doubles the number of spatial points - if we instead implement DFS as padding for a spatial-domain convolution operating in the N-S direction, we can avoid that cost increase.
> >
> > ## __Others:__
> >
> > - O1 - This is specifically referencing the fact that autoregressive error accumulation pushes trajectories away from the training distribution during long rollouts. We tackle this in two ways: first, by mitigating the known sources of nonphysical error accumulation (aliasing and geometry in this case) and second by restricting the sensitivity of the model to small perturbations (via spectral normalization).
> > - O2 - Thanks, we have done this now.
> > - O3 - Thank you for pointing this out. As we convert the linear channel mixing operation inside the nonlinearity into a true residual connection in our restructured block, we got ahead of ourselves and called both residual connections. This has now been amended and we no longer use this terminology.

---

### Author Response · Authors · 2023-08-21

We thank the reviewers for their thorough reading of our work as well as their insightful and detailed feedback. We’ve updated our submission and your reviews played a vital role in improving the clarity of our work. We would note that we still need to apply the notation updates to the appendix, but we felt it was important to have the main text updated for discussion.

One item that we feel is important to address generally is the question of audience. There was a concern that perhaps this subject was more suited to a non-ML audience. However, based on general trends in the research community as well as specific submissions recently accepted to this venue, we feel very strongly that this venue is appropriate for our work.

For context, this submission refines and provides additional insight into an approach first introduced in the original Fourier Neural Operator paper which was published in ICLR in 2021 and now has over 1000 citations. Machine learning conferences now regularly have submission categories for ML in the physical sciences - this year’s NeurIPS gave the option “Machine learning for physical sciences (for example: climate, physics)”. This year’s NeurIPS also features two separate workshops on machine learning applications in physics (ML4Physics and ML4Science) while the most recent ICML hosted a workshop on joint simulation-ML approaches. TMLR itself had featured three separate articles in the area in the month prior to our submission [1][2][3], two of which even focus on the same families of methods [2][3].

As a result, we believe that these topics are directly relevant and interesting to a core audience of this venue. In this context, the multi-disciplinary nature of our submission is something we feel is a source of excitement rather than something to be avoided. We greatly appreciate reviewer suggestions on improving or clarifying specific language which may not be as familiar to more general ML audiences, and will implement the suggested changes as described in our review-specific responses.

[1] https://openreview.net/pdf?id=EPPqt3uERT
[2] https://openreview.net/pdf?id=wNBARGxoJn
[3] https://openreview.net/pdf?id=j3oQF9coJd’

---

### Public Comment · ~Nick_McGreivy1 · 2023-12-12
**Guaranteeing stability in autoregressive models for physical systems**

Nice article. In future work, you might consider using invariant preservation to guarantee numerical stability for physical systems such as these. See https://arxiv.org/abs/2303.16110. Even if you don't know the underlying equations, you can guarantee stability if you know the system's underlying invariants.

---

### Decision · Action_Editors · 2023-10-02

**Recommendation:** Accept with minor revision

**Comment:**

The reviewers agreed that this is an interesting paper dealing with an increasingly important problem. While they initially had concerns about the accessibility of the paper to the machine learning community, the authors have addressed these in the new version.

Requested changes
1. Please incorporate the remaining changes promised to the reviewers, i.e. update the notation in the appendix.
2. Expand the caption in Figure 3 to describe each of the subfigures, and give them individual labels.
3. Update the last paragraph of Section 4.2 to refer to Figure 3 instead of Figure 8.

**Audience:**

This paper will be of interest to researchers using machine learning methods to model the dynamics of spatiotemporal systems, such as climate, in physical sciences.

**Claims And Evidence:**

The paper deals with the issue of compounding errors when performing autoregressive time-stepping using neural operators that model spatiotemporal systems defined by partial differential equations. The authors analyze the source of errors and propose several mitigation techniques. The results on two synthetic and one challenging real-world dataset show that the proposed techniques lead to lower prediction errors and increased stability.